# Breakdown of microbial networks links nutrient stress and reef coral disease

Raphaela Gracie[1], Jörg Wiedenmann[1], Phyllis Lam [2], Michael Sweet [3] & Cecilia D'Angelo [1] ✉

Coral diseases are increasing in prevalence, accelerating the global decline of tropical reefs, which threatens over 25% of marine biodiversity and vital ecosystem services for human societies. While outbreaks are frequently linked to environmental change, including heat stress, sedimentation, and reduced water quality, the mechanisms by which such factors promote disease remain poorly understood. Here we show that nutrient stress, caused by skewed seawater nitrogen-to-phosphorus (N:P) stoichiometry, promotes the onset of Black Band Disease (BBD), a common and easily recognisable syndrome that affects corals around the globe. Using *Turbinaria reniformis* as a model system, controlled laboratory experiments demonstrate that skewed N:P ratios disrupt the functional integrity of coral-associated microbial networks while favouring opportunists that exploit dysfunctional host–symbiont interactions. Disease lesion-associated microbial mats are dominated by cyanobacteria and include sulphur-metabolising bacteria, hallmarks of natural BBD communities. Strikingly, similar cyanobacterial taxa are also detected in the visually healthy coral tissue ahead of the expanding lesions, suggesting an opportunistic recruitment of disease-associated members from the resident microbiome. Global analyses of BBD outbreaks reveal that over 88% occurred in regions with skewed N:P ratios, compared with only 16% that were linked to prior heat stress. Together, our findings identify nutrient-driven microbiome destabilisation as a key pathway to coral disease, reinforcing nutrient management as a major lever for reef conservation and restoration practice.

Coral diseases, including black band disease (BBD) and other tissue-loss syndromes, are major drivers of coral mortality and reef decline worldwide[1–4]. Some, such as Stony Coral Tissue Loss Disease (SCTLD), are closely associated with specific bacterial genera[5–7], while others can be caused by individual, contagious pathogenic species such as *Vibrio coralliilyticus*[8–12]. The latter disrupts the coral–dinoflagellate symbiosis and cause bleaching and mortality[9]. In contrast, BBD, among the earliest described coral diseases, involves a complex microbial consortium[13–18]. The BBD prevalence in parts of the Great Barrier Reef

and the Caribbean has been shown to temporarily reach up to the order of 5%[4,19]. On other occasions, fewer than 1% of coral colonies on any reef area at any one time were affected by BBD, yet the susceptibility of major frame-building species greatly enhances the impact on coral communities[20]. Furthermore, since the disease occurs repeatedly and affects numerous coral species across the globe, it is an important contributor to reef degradation[20–23].

The disease is characterised by a dark microbial mat that migrates across coral colonies, killing tissue at the front and leaving bare

[1]Coral Reef Laboratory, School of Ocean and Earth Science, University of Southampton, Southampton, UK. [2]Marine Biogeochemistry, School of Ocean and Earth Science, University of Southampton, Southampton, UK. [3]Aquatic Research Facility, Nature-based Solutions Research Centre, University of Derby, Derby, UK. ✉e-mail: c.dangelo@soton.ac.uk

skeleton behind[15]. The mat community is dominated by phycoerythrin-rich, gliding, filamentous cyanobacteria that lend the band its characteristic dark pigmentation[17]. Photosynthates from the cyanobacteria and the hypo- and anoxic conditions created in the mat foster the growth of sulphate-reducers which produce toxic hydrogen sulphide, which is, together with cyanobacterial microcystins, the decisive cause of coral tissue mortality[14,24,25].

High-throughput sequencing studies have extensively characterised prokaryotic communities in BBD lesions, revealing variations in microbial composition across disease stages, geographic regions, and host coral species[26–29]. While cyanobacterial patch infections are considered a frequent precursor of BBD[14,30,31], it remains unclear why and how these patches form in the first place and if they facilitate the transmission between coral colonies in natural reef environments[14,32].

Seawater temperature is considered a key driver of BBD virulence, and elevated temperatures have been correlated with higher disease prevalence[4,16,30,32–37]. Increased dissolved inorganic nutrient concentrations, including nitrate and phosphate, have also been linked to enhanced BBD progression and severity[22,30,38].

Due to the frequent occurrence and distinct appearance of the disease, numerous observations of BBD are published (Supplementary Data 1). However, only a few studies report relevant environmental parameters preceding or accompanying disease incidences. Accordingly, it remains difficult to develop management strategies that can help to prevent future outbreaks. Exceptions include studies from the Bahamas and Florida Keys, which provide the concentration of seawater nutrients during the time of the BBD observations[30,31]. With reported values of up to ~30, the ratio of total dissolved inorganic nitrogen (N) to soluble phosphorus (P) in sites with BBD was considerably higher compared to locations without BBD (N:P ~ 22)[38]. Notably, experimental nutrient enrichment that enhanced BBD progression[22] was associated with increases in N:P ratios from ~22 to ~58 under in situ enrichment conditions and from ~20 to ≥62 in tank experiments. In the latter case, the rate of BBD progression increased as the N:P ratio rose, driven by nitrate addition to the seawater[22]. Such skewed dissolved inorganic N:P ratios can have serious detrimental effects on symbiotic reef corals[39,40], ranging from increased bleaching susceptibility[41–43] to reduced growth[44–46]. At the same time, seasonal changes in nutrient availability are considered the most important environmental factors influencing coral-associated bacterial community composition[47].

We therefore postulate that skewed N:P ratios in seawater could promote BBD by altering coral-associated microbiomes. To test this hypothesis, we examine the effects of imbalanced N:P ratios on the coral *Turbinaria reniformis* under controlled laboratory conditions. *T.*

*reniformis* is not only a well-established experimental model[47], but also a member of a genus known to be susceptible to BBD in natural reef environments, hence well suited for such disease studies[48,49]. In addition to general visual health metrics, we continually record the maximum quantum yield of PSII ($F_v/F_m$) of the coral's symbionts in vivo by pulse modulated amplitude (PAM) fluorometry as a proxy for stress[44]. Finally, we use data archives to reconstruct the temperature history and nutrient stoichiometry for regions with reported BBD outbreaks to determine the wider relevance of these parameters for disease prevalence.

## Results

### Skewed seawater N:P stoichiometry results in shifts in bacterial community composition and the onset of coral disease

Corals kept under control conditions (High Nitrate: High Phosphate; HN:HP, N:P ~ 8), remained healthy throughout the duration of the 73-day-experiment, with symbionts exhibiting no change in their maximum PSII quantum yield (Fig. 1 and Supplementary Fig. 1). In contrast, exposure to imbalanced nutrient conditions (Low Nitrate: High Phosphate; LN:HP, N:P «1, and High Nitrate: Low Phosphate, HN:LP; N:P»30) led to the development of coral tissue lesions that were surrounded by advancing dark brown microbial mats, separating bare coral skeleton from live tissue (Fig. 1A–D). These lesions closely resembled the appearance of BBD in published reports[31]. In LN:HP conditions, the maximum quantum yield of PSII of the coral symbionts did not indicate any detectable signs of stress (Supplementary Fig. 1). However, disease lesions appeared after 24 days (Fig. 1E). In contrast, in the HN:LP treatment, decreasing maximum PSII quantum yield of symbiont photosynthesis was observed (Supplementary Fig. 1), and disease lesions were present within 10 days (Fig. 1E).

We also assessed the bacterial community composition of coral areas with visually healthy tissue under control and skewed nutrient conditions ~7 weeks after the start of the experiment. Next-generation 16S rRNA amplicon sequencing identified 12,771 unique amplicon sequence variants (ASVs) across all coral colonies, including 12,730 bacterial and 41 archaeal taxa. In contrast, a total of 5699 bacterial and 121 archaeal taxa were identified in seawater samples from the different treatment tanks.

There was no difference in richness, diversity or dominance between the coral-associated communities and those in the seawater under control conditions (Fig. 2A–D). In the LN:HP treatment, all four alpha diversity metrics were significantly reduced in seawater samples (Supplementary Data 2). In HN:LP conditions, richness (ASVs and Chao1) was lower in seawater, but there were no significant differences in diversity and evenness compared to coral-associated microbes.

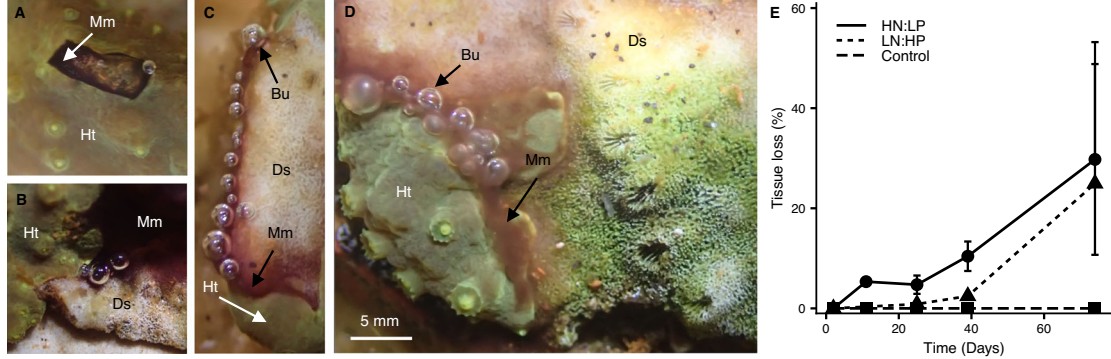

**Fig. 1 | Progression of disease lesions in *Turbinaria reniformis* under skewed N:P stoichiometry.** **A**–**D** Representative images of Black Band Disease (BBD) lesions in *T. reniformis* under nutrient-imbalanced conditions. Images show oxygen-rich bubbles attached to migrating dark-pigmented microbial mats separating healthy tissue and dead skeleton. Mm microbial mat, Ht healthy tissue, Ds dead skeleton, Bu bubbles. **E** Progressive expansion of disease lesions given as mean ± standard deviation of percentage tissue loss over time.

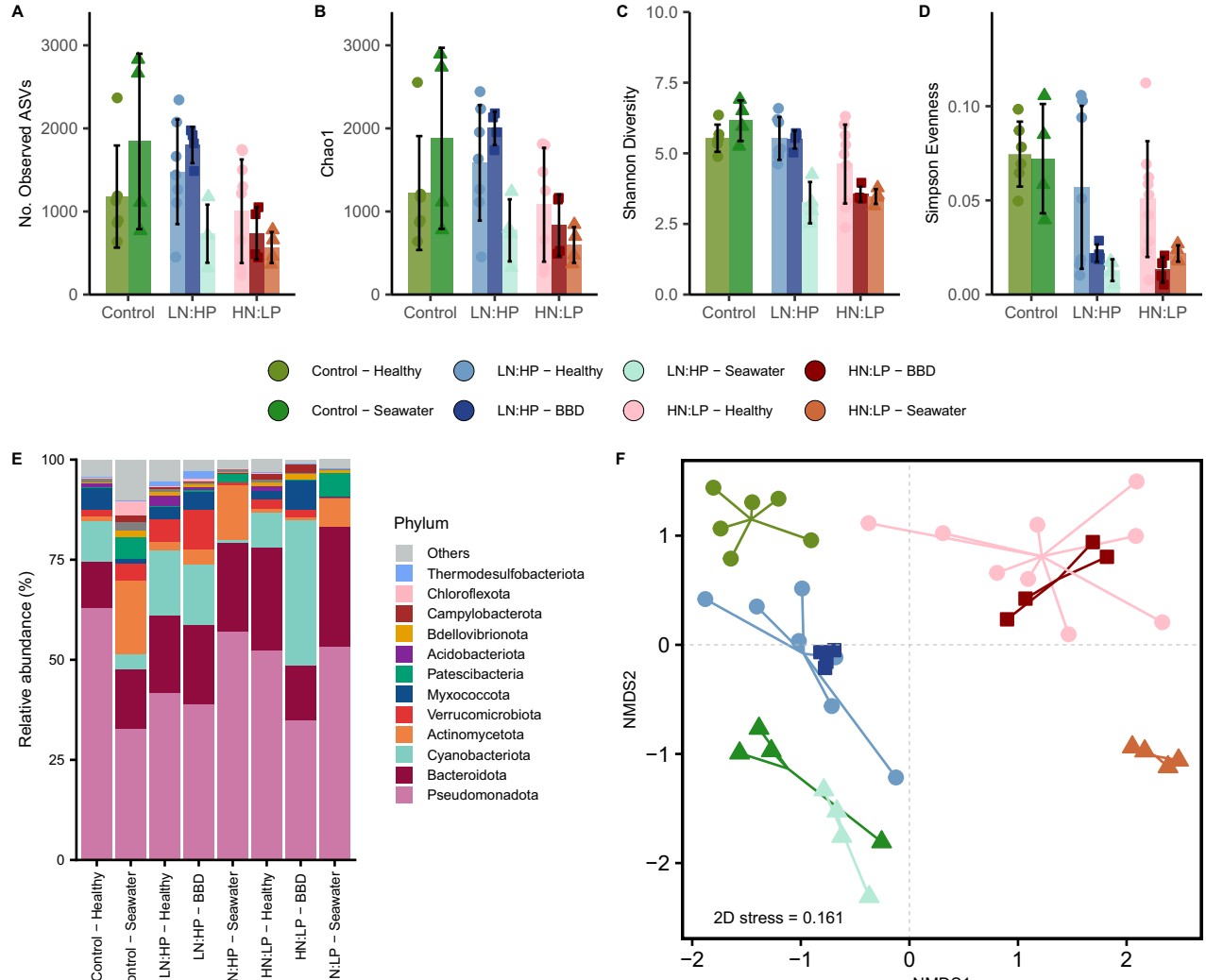

**Fig. 2 | Alpha and beta diversity of prokaryotic communities associated with *Turbinaria reniformis* and seawater under control, LN:HP and HN:LP conditions. A–D** Mean ± standard deviation of alpha diversity measures characterising the bacterial communities of corals (n = 22), seawater (n = 12) and BBD mats (n = 8). **E** Phylum-level relative abundance of prokaryotic communities found in healthy tissue, disease lesions and seawater under nutrient replete, LN:HP and HN:LP conditions. **F** NMDS ordination of Bray-Curtis distances showing beta-diversity of rarefied relative abundance ASV counts. Each data point represents a replicate coral colony (spheres), seawater sample (triangles) or disease lesion (squares). Statistical comparisons are detailed in Supplementary Data 2.

Shannon diversity and Simpson evenness were significantly lower in seawater compared to coral-associated microbial communities (Fig. 2A–D and Supplementary Data 2).

*Pseudomonadota* and *Bacteriodota* were the dominant taxa among the microbial communities of coral tissue samples and the surrounding seawater across all nutrient conditions (Fig. 2E). Interestingly, *Cyanobacteriota*, including taxa that have been previously identified in BBD microbial mats in natural reef settings, constituted on average ~11% of the total bacterial communities from healthy coral tissue areas in all nutrient environments (Fig. 2E; Supplementary Table 1; Supplementary Figs 2 and 3). In contrast, *Cyanobacteriota* were detected only in very low abundance in seawater, comprising ~1.5%.

Overall, Bray-Curtis dissimilarity measurements, visualised by non-metric multidimensional scaling (NMDS), clearly distinguish between coral-associated bacterial communities and those from seawater across the different nutrient environments (Fig. 2F). The beta diversity of bacterial communities associated with healthy coral tissue across all nutrient conditions shows homogeneity in group dispersions but reveals significantly different community compositions in each condition (Fig. 2F and Supplementary Data 2). Similarly, the beta-diversity of seawater bacterial communities from the different nutrient environments shows homogeneity in group dispersions whilst having significantly different community compositions (Supplementary Data 2).

Compared to the control corals, exposure to skewed nutrient ratios resulted in a 10 to 32 times increase in the abundance of specific *Cyanobacteriota*. However, the key taxa were distinct across conditions (Supplementary Fig 2). *Cyanobacteriota* were also among the most relatively abundant taxa of the microbial mats from disease lesions, representing ~15% of the mat communities in corals from LN:HP and ~37% in HN:LP conditions, respectively (Fig. 2E, Supplementary Fig. 3A). Phylogenetic analysis of all cyanobacterial sequences from disease lesions across treatments revealed clusters of similar taxa. Many of them were identical or closely related to cyanobacteria from natural BBD mat consortia (Supplementary Table 1; Supplementary Fig. 3A).

Notably, microbial mats showed a highly similar alpha diversity compared to healthy tissue areas (Fig. 2C and Supplementary Data 2). They also clustered with those from healthy parts of the same corals in NMDS plots (Fig. 2F). However, significantly lower species richness

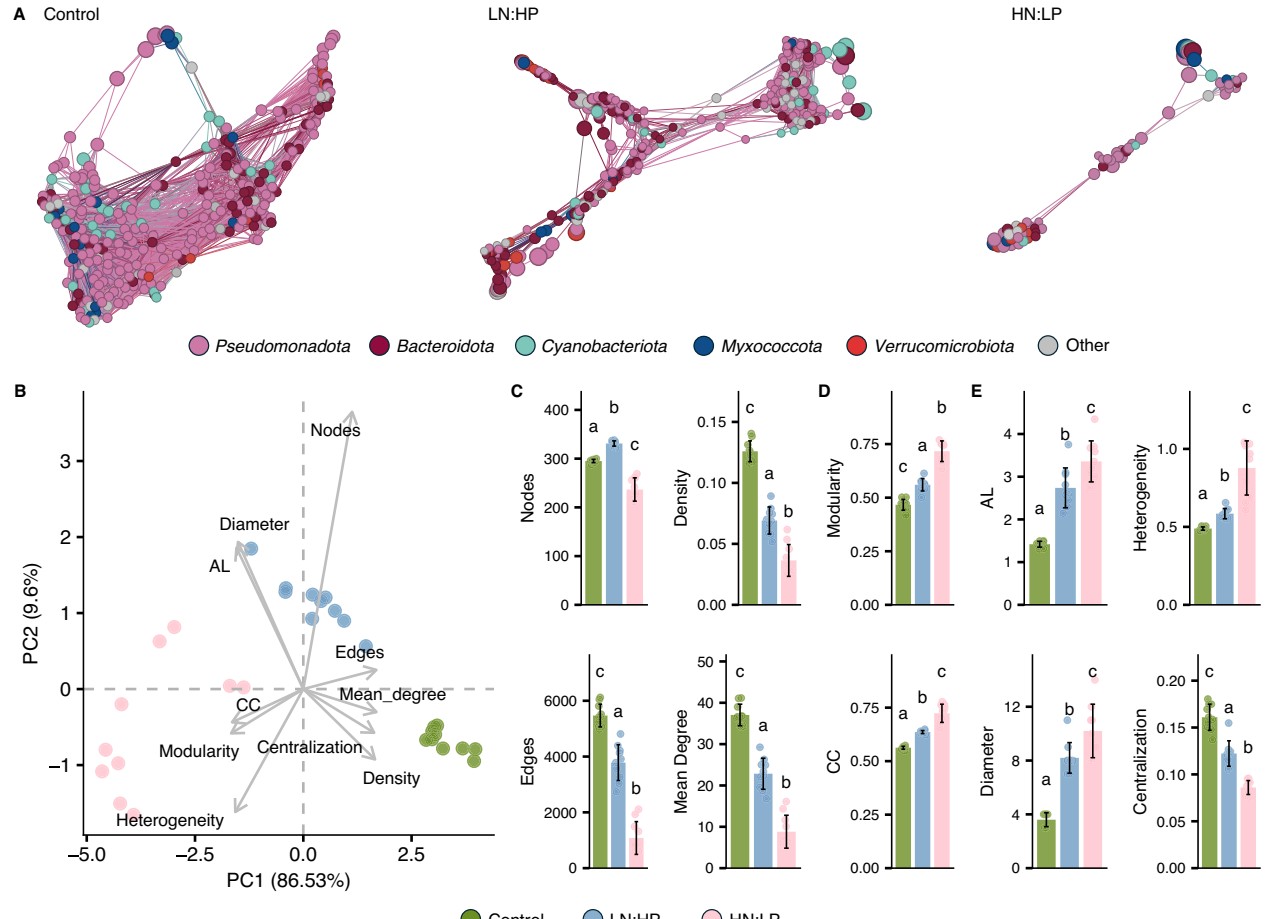

**Fig. 3 | Network topological properties of coral-associated prokaryotic communities. A** Shows the hypothetical community networks based on Bray-Curtis dissimilarity at the ASV level. **B**–**E** Show network properties for *Turbinaria reniformis* colonies under control, LN:HP, and HN:LP conditions. **C** Shows the network properties relating to connectivity and size (nodes, edges, density and mean degree). **D** Shows the properties relating to network structure (modularity; CC, clustering co-efficient), and **E** shows the properties relating to network efficiency and resilience (AL, average shortest path length; diameter; heterogeneity and centralisation). Network topological properties were calculated from replicate ($n = 10$) networks constructed using random community subsamples for each nutrient condition. Bars represent mean ± standard deviation. Significant differences ($p \leq 0.05$) between nutrient treatments were determined by a one-way ANOVA. Post-hoc comparisons were conducted using Tukey's HSD for homoscedastic data and Games-Howell where variances were unequal. Treatments with different letters are significantly different. Statistical results are shown in Supplementary Data 3.

(observed taxa and Chao1) and Shannon diversity were identified in disease lesion communities under HN:LP conditions compared to those from LN:HP conditions (Fig. 2A–C and Supplementary Data 2).

Finally, a relative abundance of sulphur-metabolising, anaerobic bacteria (*Thermodesulfobacteriota*) of up to 3% of the total microbial mat population was determined in the BBD lesions (Fig. 2E, Supplementary Fig 3B and Supplementary Table 1B).

Overall, taxonomic alignment (Supplementary Table 1 and Supplementary Fig. 3) demonstrates a high level of similarity between prokaryotes identified in this study and BBD-associated taxa reported from field samples, spanning key functional groups involved in BBD initiation and progression, namely (1) *Cyanobacteriota*, (2) sulphate-reducing bacteria, (3) sulphide-oxidising bacteria, and (4) heterotrophic or opportunistic taxa.

## Skewed N:P stoichiometry disrupts the structure of coral-associated bacterial communities

Co-occurrence network analyses demonstrate that a major restructuring of bacterial communities occurred in the visually healthy tissue when exposed to skewed N:P ratios. Under both LN:HP and HN:LP conditions, the community networks show an increased modularity with reduced connections across individual taxa (Fig. 3A). The analysis

of topological properties by principal component analysis reveals a clear grouping of the bacterial communities in response to each of the nutrient environments tested in our study (Fig. 3B). A significant decrease of metrics characterising the connectivity and size of the networks was measured for bacterial communities of corals exposed to either LN:HP or HN:LP treatments (Fig. 3C and Supplementary Data 3). These communities also showed a significant increase in modularity and clustering coefficients, indicating a loss of network structure (Supplementary Data 3 and Fig. 3D).

Average path length, diameter and heterogeneity of bacterial networks associated with corals exposed to skewed N:P ratios increased, while network centralisation was reduced (Fig. 4E and Supplementary Data 3). These proxies indicate more fragmented and potentially less cooperative and resilient microbial communities[50]. These effects were most pronounced in samples from HN:LP conditions. Specifically, the network size was reduced by ~80% (HN:LP) and ~30% (LN:HP) compared to the controls.

## Temperature history and N:P stoichiometry of BBD-affected regions across the globe

We used global data repositories to reconstruct the temperature history and N:P stoichiometry of BBD-affected regions that were specified

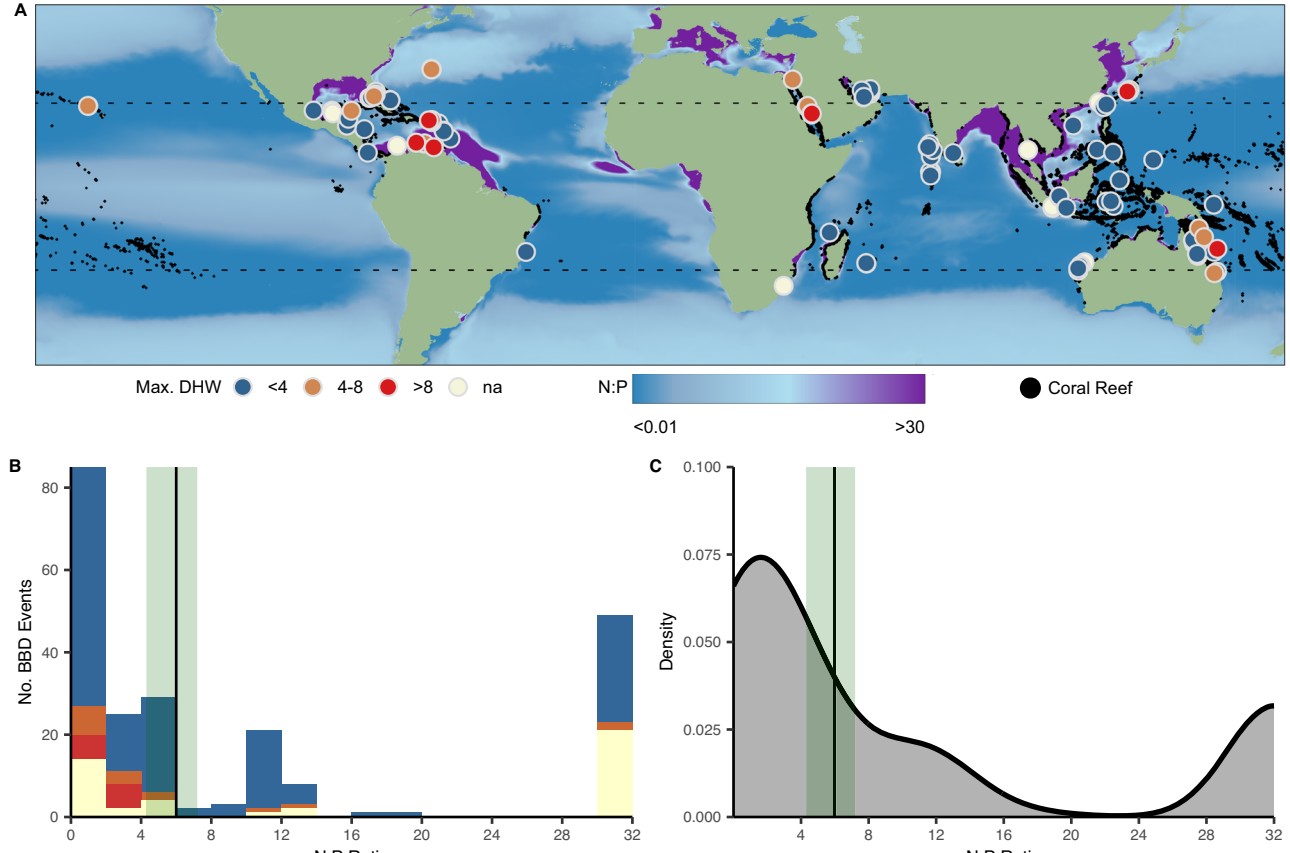

**Fig. 4 | Global distribution of black band disease between 2006 and 2023.**
**A** Locations of coral colonies identified with Black Band Disease (BBD) reported in published research papers (n = 230 unique sites and timepoints, or incidence points, where BBD was reported across n = 141 primary research papers). Data point colour shows maximum degree heating week (DHW) for each BBD incidence point in the previous year. Data points are ranked so that BBD incidence points with higher DHW values are plotted on top of those with lower DHW values. Colour ramp shows the prevailing global surface seawater N:P stoichiometry, derived from Bio-ORACLE 3.0, at a spatial resolution of 0.05°. **B** Frequency and **C** Kernel density estimates of BBD incidence points according to the prevailing reef seawater N:P stoichiometry. Green shading indicates the typical range of N:P ratio (4.3–7.2) for nitrogen-limited reef waters with solid black lines marking an average N:P ratio of ~6.

in reports published between 2000 and 2023. Mapping of locations where BBD was reported confirms the global distribution of the disease (Supplementary Data 1 and Fig. 4A).

We first used satellite-derived temperature data from the NOAA Coral Watch Program to calculate degree heating weeks (DHW)[51] during the 12 months preceding local BBD outbreaks. DHW values > 4 were considered to be indicative of coral stress, as this threshold is employed by the NOAA Coral Watch Program as a predictor of possible bleaching. DHW values > 4 were associated with only 16% of the reported BBD events (Fig. 4A; Supplementary Data 1 and 4). Regional long-term mean surface nitrate (N) and phosphate (P) concentrations were retrieved from BIO-ORACLE v3.0 for a fixed decadal period (2000–2010)[52] to characterise the N:P stoichiometry for sites affected by BBD outbreaks. We considered reef waters to be N-limited if the dissolved inorganic N:P ratio is smaller than the canonical Redfield N:P ratio of 16[53], specifically if it falls in or below the range of ratios between ~4 and ~7, previously reported for different reef environments[54–56]. Ratios between ~4 and ~7 were set as a boundary range to identify conditions with deviating stoichiometry[41]. We found that just 12% of the BBD incidences occurred in nitrogen-limited reefs with N:P ratios in ~4–7 (Fig. 4B, C). In contrast, the majority (88%) of BBD reports were from sites characterised by strongly skewed N:P ratios below 4 or above 30 (Fig. 4B, C).

Across the 24-year study interval, 36 incidences of BBD were reported off the southeast coast of Florida and the Florida Keys. Of the 15 BBD incidences for which both location and time of identification were reported, prior heat stress exposure (DHW = 5) was detected in only two cases (Supplementary Fig. 4A). Instead, the whole affected region is characterised by N:P ratios of 30–560, with land-derived nutrients primarily delivered through the Everglades, including the Shark River Slough[57] (Supplementary Fig. 5B). The nutrient introduction into coastal waters increases with the amount of runoff and riverine discharge[58,59]. The daily discharge volume, and thereby nutrient influx, from the Shark River Slough was above the long-term average for an average of 47 days in the years with reported BBD outbreaks (Supplementary Fig. 4C).

## Discussion

We have combined controlled laboratory experiments with a global analysis of field observations to test the hypothesis that BBD can be promoted by skewed dissolved inorganic nitrogen-to-phosphorus (N:P) ratios. We showed that the microbiome of the model coral *Turbinaria reniformis* is distinct from the surrounding seawater and responds specifically to nutrient conditions. This mirrors observations from natural reef settings, where coral-associated microbial communities show strong sensitivity to nutrient availability[47].

Across all tested nutrient conditions, the dominant coral-associated taxa remained stable and included groups such as *Pseudomonadota* and *Bacteriodota*, a trend commonly observed in field-derived coral microbiomes[60]. *Cyanobacteriota*−central players in the

formation of BBD mats[31]—were significantly enriched even under balanced nutrient conditions and in intact coral tissue compared to the seawater. Several representatives were identical or closely related to taxa previously identified in BBD lesions in the field[24,29,61–69]. This is further underpinned by a high-resolution taxonomic and phylogenetic similarity between sulphate-reducing bacteria found in our experiments and established taxa involved in BBD disease aetiology[70].

In treatments with skewed N:P ratios, cyanobacteria also dominated the microbial assemblages of mats that developed on and around tissue lesions. Key taxa matched those detected in intact tissue under controlled conditions, suggesting that *Cyanobacteriota* can expand opportunistically to establish mats.

Anaerobic, sulphur-metabolising bacteria, identified as likely secondary pathogens in BBD, are known to increase the virulence of infection by creating a sulphide-rich, anoxic microenvironment that kills the coral tissue underneath the microbial mat[30,31]. Our analyses show that *Thermodesulfobacterota* accounted for up to 3% of the total microbial mat community. Members of this anaerobic group are capable of reducing sulphate, as well as elemental sulphur, to sulphide[71]. Comparable proportions of sulphate-reducing bacteria (1–7.5%) have also been reported in environmental BBD samples[25].

In our experiments, the microbial mats spread outwards from initial starting patches, overgrowing and killing the underlying coral tissue[31]. This supports earlier suggestions that BBD likely originates from small cyanobacterial patch infections before progressing to a band[14].

The BBD microbial communities experienced a reduction of diversity under both HN:LP and LN:HP conditions. Interestingly, their similarity to the healthy tissue communities from the same treatment was greater than their similarity to each other. This phenomenon has been previously observed in other coral tissue-loss diseases[72]. Nutrient imbalance also induced major restructuring of microbial co-occurrence networks, with a marked reduction in overall complexity. Such simplification has been linked to a loss of resilience and functional redundancy in other microbial systems[73,74]. In corals, reduced network functionality may weaken the beneficial, probiotic roles of associated microbes[75], facilitating dominance shifts and allowing invasive or opportunistic taxa to proliferate.

Analysis of the temperature history and N:P stoichiometry of BBD-affected reef regions across the globe indicates that disease outbreaks are promoted by N:P ratios below or well above ~6, an average value reported for nitrogen-limited reef environments[41,54–56,76]. By contrast, thermal stress was associated with only a minority of outbreaks. While elevated temperatures may still exacerbate disease onset and progression[4], our work indicates that nutrient stoichiometry is a strong determinant of BBD prevalence.

Under strongly nitrogen-limited conditions (N:P < 4), cyanobacteria likely gain a selective advantage through their ability to fix nitrogen[77]. Nitrogen fixation is energetically costly, requiring abundant energy-rich compounds[78]. These can be derived from their own photosynthetic activity but may also be supplemented by coral-derived carbon compounds. Under strong N-limitation, the corals' symbiotic dinoflagellates release most of their fixed carbon to the coral host, as they cannot use it for their own growth[79]. This is supported by the consistent finding, demonstrated here and in previous work, that the functional integrity of the symbiont photosynthetic machinery is maintained even under pronounced nitrogen limitation[41,42,45]. The host itself, equally growth-limited under these conditions, may subsequently release excess carbon compounds, for instance, in the form of polysaccharide-rich mucus[80]. Thereby, this surplus carbon becomes accessible to coral-associated microbes, including cyanobacteria. Once the cyanobacterial mats are established, the hypoxic or anoxic conditions within them likely recruit secondary pathogens in the disease consortium, such as sulphur-metabolising bacteria[30,31,68]. Future research should test the rates of nitrogen fixation[81] in different nutrient environments to provide a quantitative framework for this potential mechanism.

Conversely, unusually high N:P ratios (>30) also promote BBD outbreaks. Under phosphorus starvation, the nitrogen-fixation capacity of cyanobacteria becomes limited[77]. In addition, the maximum quantum yield of PSII of the corals' algal symbionts declines, a trend consistently found in our present experiments and in earlier studies[40–42], affecting the availability of energy sources for nitrogen fixers. In this scenario, photosynthetic cyanobacteria may gain a competitive advantage by exploiting phosphorus released from compromised coral tissue to maintain the productivity of their own photosynthetic machinery. Localised injuries, such as fish bites or mechanical damage, could provide access to phosphorus pools, fostering the establishment of cyanobacterial mats as BBD precursors[14,30].

Taken together, BBD presents as an opportunistic infection[26,82], shaped by environmental conditions, in particular by seawater N:P stoichiometry. Specifically, nutrient imbalance disrupts the functional integrity of coral-associated microbiomes while simultaneously favouring opportunists that exploit dysfunctional host-symbiont relations. Several cyanobacterial taxa from the BBD mats of our experimental corals, though previously isolated from the environment, were undetected or present only in extremely low abundance, in our seawater samples. Instead, they were associated with visually healthy coral tissue, indicating that the host microbiome is a key reservoir for opportunistic pathogens under nutrient-imbalanced conditions. Comparable processes occur in humans, where opportunistic infections, such as thrush, following antibiotic-induced microbiome disruption[83] or the bacterial invasion of virus-damaged lung tissue, arise from environmental imbalances and altered host-microbiome interactions rather than from the direct infection with single, contagious pathogens[84,85]. Future research should test the effects of nutrient stoichiometry on additional coral species and consider also other environmental stressors that could potentially trigger microbial imbalance causing BBD.

Nutrient perturbations, including altered N:P ratios, are nowadays recognised as key drivers of coral reef decline[40,41,43,46,86,87]. Our results suggest that restoring natural nutrient stoichiometry may help prevent BBD outbreaks. The balance of N:P, rather than absolute nutrient load, appears most critical in this context. Nonetheless, nutrient enrichment broadly undermines the competitiveness of symbiotic reef corals, which depends on nutrient-poor conditions[36,81]. Preventing any form of anthropogenic nutrient inputs into reef waters must therefore remain a central goal of coastal management strategies.

## Methods

### Aquarium experiments

Coral colonies were cultured and propagated by fragmentation to produce genetically identical replicate fragments or ramets in the experimental mesocosm of The Coral Reef Laboratory, University of Southampton[88]. Coral animal husbandry was performed in compliance with the relevant ethical regulations at the University of Southampton. Coral fragments were allowed to recover for >14 days in optimal growing conditions before random distribution into experimental systems.

All coral culture experiments were conducted across three experimental systems, each with a defined seawater N:P stoichiometry. The terms 'high' and 'low' describe the relative concentrations of $NO_3^-$ and $PO_4^{3-}$ within each experimental system as described previously[41,42,45]. Nutrient conditions with high nitrate/high phosphate (HN:HP, $NO_3^-$ ~15.2 μmol/L, $PO_4^{3-}$ ~3.55 μmol/L, N:P ~ 8) were used as control, simulating nutrient environments that have previously been described for high nutrient reef environments (Galápagos Islands, Brazilian coast) or in reefs with internal wave-driven upwelling[76,89,90]. In our experimental aquarium system, these conditions promote

vigorous coral growth[91]. Ammonium concentrations in the experimental systems were constantly low, therefore the measured $NO_3^-$ represents a majority of the total dissolved inorganic nitrogen pool[42]. Two nutrient imbalanced conditions were defined by the N:P ratio as high nitrate/low phosphate (HN:LP, N:P > 30) and high phosphate/low nitrate (LN:HP N:P < 4). Nitrate was removed continuously from the LN:HP system using a Nitrate Reactor (Aqua Medic). Phosphate concentrations were kept at low levels in the HN:LP system by recirculation through a Rowaphos phosphate-removal matrix (D-D The Aquarium Solution). If needed, macronutrient levels were increased by dosing sodium nitrate or sodium phosphate solutions[88]. Experimental systems were supplemented with iron and other trace elements through regular dosage with commercially available solutions (Coral Colours, Red Sea), in addition to weekly water changes with artificial seawater using Pro-Reef salt mixture in demineralised seawater (Tropic Marin)[41]. Corals were cultured at $26 \pm 1\,°C$, salinity $33 \pm 0.5$, and a photon flux of ~120 $\mu mol\ m^{-2}\ s^{-1}$ under a 12:12 h light/dark cycle[32,34]. The light regime was selected based on light levels in the natural habitats in 4 m depth of the closely related species *T. mesenterina* and on evidence that optimal growth irradiances for this species occur toward the lower end of its habitable light intensity range, peaking at approximately 300 $\mu mol\ m^{-2}\ s^{-1}$ [92]. The lighting, heating, water movement, filtration and macronutrient concentration parameters for the experimental system have been described previously[34,78].

A single mother colony of *T. reniformis* was fragmented and the fragments were grown into replicate colonies for >6 months. The replicate coral colonies ($n = 18$) were then cultured under the three different seawater N:P stoichiometry systems for >7 weeks.

Coral fragments were photographed and maximum quantum yield of PSII monitored frequently throughout the experiment, approximately every week. Three replicate coral fragments were randomly selected from each experimental treatment and sampled for prokaryotic community analysis in healthy tissue areas after at least 7 weeks of treatment on days 50 ($n = 9$) and 73 ($n = 9$). Microbial lesion samples ($n = 8$) were collected on day 60 from HN:LP and LN:HP conditions. For comparison, samples from the remaining healthy tissue of the same colonies were also collected on day 60 from LN:HP ($n = 1$) and HN:LP ($n = 3$). Replicate seawater samples were collected at each sampling timepoint (days 50, 60 and 73; $n = 12$).

## Physiological monitoring

Photosystem II (PSII) maximum quantum yield (Fv/Fm) was measured after ≥10 h of dark acclimation using a submersible pulse-amplitude modulated fluorometer (Diving-PAM) (Walz) as proxy of the overall health of Symbiodiniaceae endosymbionts[44]. Declining Fv/Fm values were considered as indicative of stress experienced by the symbionts[41,45]. This PAM instrument excites fluorescence by pulse modulated red light from a light-emitting-diode (LED, 655 nm). When recording Fv/Fm measurements of the coral tissue, we ensured that the measuring area of the PAM was located at least 3 mm away from microbial mats in the proximity to prevent a spectral contamination by fluorescence signals from the microbes. The extent of tissue loss was quantified using photos of coral fragments taken from above. The area of tissue loss was calculated using the polygon tool in ImageJ (version 1.53).

## DNA extraction, sequencing and analysis

Two different methods for extracting DNA were used depending on the type of sample. DNA from microbial cells present in *T. reniformis* was extracted using a lysis solution (0.5% SDS, 50 mM NaOH and 5 nM EDTA). Tissue was homogenised using a pestle andvortex mixing. DNA was washed with butanol, precipitated with ethanol and resuspended in a 50 $\mu l$ Tris buffer (10 mM Tris-HCl pH 8.5). Microbial DNA from seawater and bacterial mat samples was extracted using the Qiagen DNeasy PowerWater Sterivex kit.

Preparation of 16S rRNA gene amplicons and sequencing using the Illumina MiSeq sequencer was carried out at the Environmental Sequencing Facility, University of Southampton. Amplification of the V3-4 region of the 16S rRNA gene was conducted using the illumina-pro341F [5′TCGTCGGCAGCGTCAGATGTGTATAAGAGACAGCCTACGGGNBGCA SCAG3′] and illumina-pro805R [5′ GTCTCGTGGGCTCGGA-GATGTGTATAAGAGACAGGACTACN VGGGTATCTAATCC3′] primer pair. Primer sequences contained sequencing adaptor overhangs that are underlined. Amplicon PCR reactions were run using NEBNext® Ultra™ II Q5® Master Mix with 5 ng/$\mu l$ DNA, 1 $\mu M$ illumina-pro341F forward primer and 1 $\mu M$ pro805R reverse primer followed by PCR clean-up using AMPure XP beads to purify amplicon sequences from primers and primer dimer species. Illumina sequencing adaptors and dual indices were attached using the Nextera XT Index Kit followed by a second PCR clean-up using AMPure XP beads. The final DNA library was quantified, normalised and pooled. Pooled libraries were denatured with NaOH, diluted with hybridisation buffer and heat denatured before MiSeq sequencing with reagent kit v3 and a 5% PhiX as spike-in.

All 16S rRNA amplicon sequencing data were processed using R version 4.4.2 and RStudio 2025.05.1 + 513[93]. The analytical pipelines included: (i) Filtering and trimming: The quality of forward and reverse reads was assessed using the DADA2 pipeline[94]. Reads were truncated to remove primer sequences and low-quality bases at the 3′ end. Reads containing N values or more than two expected errors were removed from the analysis[95]. (ii) Dereplication and denoising: Identical reads were clustered before using the DADA2 machine learning algorithm to apply a parametric error model to learn the sequencing error rates within each dataset. A model of sequencing errors was constructed for each dataset to account for any variation between sequencing runs. Error models were used for denoising and inferring ASVs. Chimeric reads and ASVs with a total abundance of <10 across all samples were removed from further analyses. (iii) Assigning taxonomy: Taxonomic classification was assigned to ASVs using DECIPHER[96] and the SILVA SSU r138 (version 2, 2024) database[97]. Mitochondrial and chloroplast sequences were removed after taxonomic classification. (iv) Rarefaction: All samples were randomly subsampled and rarefied to account for uneven sequencing depths among samples[98]. Construction of rarefaction curves showed that samples were reaching saturation, indicating sufficient sampling depths.

Analysis of compositions of microbiomes with bias correction (ANCOM-BC2)[99] was used for differential abundance analysis of unrarefied ASV counts data. ANCOM-BC2 estimates the unknown sampling fractions and corrects the bias induced by differences among samples. Differentially abundant taxa are identified while controlling the FDR for multiple testing. ANCOM-BC2 also provides 95% simultaneous confidence intervals for the mean differential abundance of each taxon between each pair of experimental groups. The absolute abundance data are modelled using a linear regression framework[100]. For analyses using ANCOM-BC2, $p$-adjusted values controlled for false detection rate, filtered for sensitivity analysis and a conservative variance estimator for the test statistic was used due to the small sample size. Results are represented by effect size (log fold change).

Co-abundance networks were constructed from prokaryotic community data using Bray-Curtis similarity to assess the overall structure of prokaryotic communities[72,101,102]. A prevalence filter was applied to remove zero-rich taxa[103]. Sequencing depth and evenness are confounding factors that can influence network construction. As such, each group was normalised to account for uneven sequencing depth, and community evenness was analysed using alpha diversity metrics. No significant differences in alpha diversity measures were identified between nutrient conditions in corals, and all samples were rarefied to equal sampling depth. Network topological properties, such as network size, modularity, average degree, and number of connections (known as edges), can be used to infer microbial community connectivity and complexity and identify potential biologically

relevant microbial interactions. Additionally, network modules identify closely connected sub-communities that may indicate microbial niches within the habitat[103,104].

Cyanobacterial 16S rRNA amplicon sequences were aligned using the Multiple Alignment using Fast Fourier Transform (MAFFT) alignment program and a maximum likelihood phylogenetic tree was estimated with IG-TREE (v3.0) using 1000 ultrafast bootstrap replicates to assess nodal support. Sequences outside of those generated in this study were obtained from the NCBI GenBank database.

## Literature analysis
A literature search was conducted using Web of Science. A search for research papers published in the 24-year period between 2000 and 2023 including 'black band disease' in either the title or abstract was conducted. The cut-off in the year 2000 was introduced based on the availability of corresponding nutrient data from the Bio-Oracle database. Each paper was downloaded and reviewed individually. Non-relevant papers were not considered further. Next, each paper was searched for the following terms: 'nutrient', 'nitrate', 'phosphate' and 'temperature'. Any paper with a latitude and longitude or map locating the diseased corals was selected and location information was collated and mapped in QGIS (v3.4). A total of 141 studies were included in this analysis (Supplementary Data 1).

## Satellite data analysis
The NOAA Coral Reef Watch (CRW) daily 5-km satellite coral bleaching Degree Heating Week (DHW) product[51] (version 3.1, released August 1, 2018) was used to reconstruct the thermal history of reported BBD events. The Degree Heating Weeks (DHW) data show the accumulated heating levels for a specific location with 5 km resolution. We used DHW data for the 365 days prior to a BBD event as metric for anomalously high temperatures experienced by each BBD incidence point (Supplementary Data 4). Next, global N:P seawater stoichiometry was calculated from nitrate and phosphate surface seawater (0–0.49 m depth) concentrations obtained from Bio-ORACLE v3.0[52,105] using QGIS and the prevailing N:P conditions were extracted for each BBD incidence point (Supplementary Data 4). Bio-Oracle does not provide ammonium and nitrite values, which contribute to the total amount of dissolved inorganic nitrogen (DIN). Accordingly, the biological relevant N:P ratios reported here are based on nitrate data only and may be lower compared to the total DIN based ratios experienced by the corals. In the context of the provided orders of magnitude, this does not necessarily make a substantial difference: Bio-Oracle derived nitrate-based data sets returned N:P ratios of ~30 to ~560 for the Florida Keys region for the period 2000–2018, whereas a dedicated nutrient survey in this region for the years 1995–98 reported median total DIN:P ratios in the range of 32–160[106].

## Shark River Slough discharge data analysis
Water flow discharge data were obtained from the US Geological Survey's (USGS) National Water Information System (NWIS). We used data from the gaging station located at Shark River Below Gunboat Island near Flamingo, Florida (USGS site no. 252230081021300) to quantify water flux from the Everglades. We used 20 years of daily discharge data from this location to calculate the long-term monthly mean water flow. Discharge data for the 365 days prior to BBD being reported at each BBD incidence point in the Florida region were collected. Daily discharge data 1 standard deviation higher or lower than the long-term monthly mean was categorised as 'high' or 'low', respectively. We note in this context that the Everglades run-off contributes only a part of the nutrients that are delivered from the wider region to these reefs through oceanographic processes[106–109]. Hence, it is not possible to directly link the unusually high N:P ratio of the region[86,106] to the nutrient enrichment through coastal processes alone[39].

## Statistics and reproducibility
Details on the number of animals, samples and analytical designs are provided in the individual sections above. Sample numbers and relevant statistical comparisons are also included in the corresponding figure legends. Outputs of statistical analysis are detailed in Supplementary Data 2 and 3.

## Reporting summary
Further information on research design is available in the Nature Portfolio Reporting Summary linked to this article.

## Data availability
The complete list of primary research papers utilised in this study; output details of statistical analyses; and compilation of temperature and nutrient data retrieved from open access databases are provided in Supplementary Data 1–4. 16S rRNA sequencing count tables and taxonomy, Fv/Fm, and tissue loss measurements data supporting the findings of this study are available via the PURE repository at the University of Southampton UK, subject to standard CC-BY license terms, and can be accessed through the link: https://doi.org/10.5258/SOTON/D3751. Raw 16S rRNA amplicon sequence data have been deposited in the NCBI Sequence Read Archive (SRA) with BioProject ID PRJNA1436489. Background land maps were generated using free vector and raster map data available at naturalearthdata.com.

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

## Acknowledgements

We thank George Clarke for his assistance with coral husbandry and continued support in maintaining the experimental mesocosm. This work was supported by the Leverhulme Trust (Research Leadership Grant RL-2022-043 to CDA) and the Natural Environment Research Council (INSPIRE DTP Scholarship to RG/CDA).

## Author contributions

C.D.A., J.W. and R.G. designed the research. R.G., C.D.A. and J.W. carried out the experimental work and data analysis. M.S. and P.L. contributed to interpretation and discussions. R.G., C.D.A. and J.W. wrote the paper with input from M.S. All authors approved the final manuscript.

## Competing interests

The authors declare no competing interests.
