## [Transparent Peer Review file · Nature Communications]

Breakdown of microbial networks links nutrient stress and reef coral disease

Corresponding Author: Dr Cecilia D'Angelo

Version 0:

Reviewer comments:

Reviewer #1

(Remarks to the Author)

This is a well-conducted study that advances our understanding of the mechanisms underlying coral disease, specifically the black band disease (BBD). It provides critical insights into how environmental factors, specifically nutrient imbalances, destabilize coral-associated microbiomes and promote disease. It fills a gap in understanding the mechanisms behind black band disease (BBD) outbreaks. The lab experiments are clearly groundbreaking and demonstrated that skewed nitrogen-to-phosphorus (N:P) ratios in seawater significantly contribute to the onset of BBD in corals. The analysis of the coral-associated microbiome from the lab experiments clearly showed disruption of the microbial network, its structure, and connectivity, in the disease tissue. Skewed N:P ratios increase modularity and reduce connections between microbial taxa, leading to fragmented and less resilient microbial communities. The global analysis of BBD outbreaks in relation to temperature and nutrient data strongly supports the conclusions. The worldwide analysis was consistent with the results from the laboratory experiment, revealing 87% of BBD outbreaks occurred in regions with skewed N:P ratios. In comparison, only 13% were linked to prior heat stress. This suggests that nutrient stoichiometry is a stronger determinant of BBD prevalence than thermal stress.

However, I am missing a more thorough discussion and presentation of the bacterial taxa detected in the microbial mats from the experiments. I don't think that the Phylum-level taxonomic resolution is sufficient to claim similarity to the other taxa commonly found in BBD worldwide. I would recommend genus-level taxonomic presentation when possible, or at minimum Order-level resolution. I would also recommend mentioning that the conclusions are based on one coral species, and testing the effects of nutrient stoichiometry on other coral species beyond *Turbinaria reniformis* would provide broader validation. I think it's possible that other environmental stressors could trigger microbial imbalance in different coral species, resulting in a similar disease presentation. Minor revisions to address confounding factors and expand the discussion would strengthen the paper.

Reviewer #2

(Remarks to the Author)

Review of Gracie et al. NCOMMS-25-84952

This is an interesting manuscript describing the relationship between nutrient stoichiometry and susceptibility to coral disease and mortality. The data available to our community, but too often overlooked, clearly shows that the majority of "coral diseases" are secondary to some sort of environmental insult (e.g., eutrophication, temperature stress, ocean acidification, etc.). Here, the authors extend their previous studies to examine the effects of deviations in nutrient stoichiometry on coral diseases. The authors use what is well described disease, with a primary pathogenic consortium of microbes, to demonstrate the role of nutrient stoichiometry in the initiation and pathogenesis of this disease. My specific concerns can be found below.

Line 41; First, While *V. coralliilyticus* appears to be a good example of temperature dependent virulence increasing the prevalence of bacterially mediated coral bleaching, *Vibrio shilonii* and *V. mediterranei* are synonymous, so *V. shilonii* doesn't even exist as *V. mediterranei* has priority from a taxonomic perspective. Second, since 2004, *V. shilonii* has no longer been detectable in either healthy or bleached corals using culture methods, fluorescent in situ hybridization (FISH), or

fluorescence microscopy using specific anti-*V. shilonii* antibodies. See Ainsworth et al. 2008, ISME J, 2:67-73 which also questions whether *V. shilonii* was ever the cause of bacterial bleaching.

Line 43; History is important. Antonious (1973) and Garrett and Ducklow (1975) were the first to describe BBD in corals.

Line 45; I see nothing in Sato et al. or in Aronson et al., quoted by Sato, to support the statement that BBD is a major source of coral mortality in the Caribbean. Dinsdale (200) reported a disease prevalence of 1.3% to 4.9% in the Caribbean. Edmunds (1991) reported prevalences as low as 0.2%.

Line 49; The most common description of this polymicrobial disease is that while the filamentous bacteria are the primary invader, it is the photosynthate from these cyanobacteria and the hypoxic/anoxic conditions created in the mat that fosters the growth of sulfate reducers with their production of hydrogen sulfide, as well as cyanobacterial microcystins, that is the decisive cause of coral mortality.

Line 57; Borger (2004), reference 29, studied primarily DSS and provided no analysis of temperature versus the prevalence of BBD.

Line 59; Sato et al. (2016), a review, does cover the role of nutrients in BBD, but there is no work on nutrients within Muller and van Woerik (2011). Voss et al. (2007) showed no effect of nutrient addition on BBD communities and Voss and Richardson (2006) used profound deviations in nutrient stoichiometry in their experimental treatment over time ending with an N:P of ~58 which you probably could use to make your case. Also, Kuta and Richardson (2002) could/should be added and replace others that are not relevant as suggested above.

Line 69; I don't see any nutrient concentration measurements from actual coral reef waters in Rosett et al. (2017), reference 33, as suggested by the text. Use the original, primary, references where those values came from.

Line 71; Lapointe et al. (2019) does discuss, extensively, the relationship between altered nutrient stoichiometry and coral bleaching, but not for coral growth, only for macroalgae. Also, Lesser (2021) discusses at length a number of issues and misinterpretations of how often both eutrophication, and deviations in nutrient stoichiometry, actually occurred in the Florida Keys during the 30 y time period reported by Lapointe et al. (2019).

Jump to M+M

Lines 295-296; Please report the exact number of genets and ramets used in the experiment, and how they were randomized in the experiment. Was genet considered as a factor in the experiment? If not, why not?

Line 320; What is the ecological context for using irradiances of 120 $\mu\text{mol quanta m}^{-2} \text{ s}^{-1}$? At a depth of 15 m irradiances can vary from ~500-1000 $\mu\text{mol quanta m}^{-2} \text{ s}^{-1}$ on many Caribbean and Pacific coral reefs (see Lesser 2024). An irradiance of 120 $\mu\text{mol quanta m}^{-2} \text{ s}^{-1}$ is therefore occurring at deeper depths where it has been noted that BBD actually declines in prevalence.

Lines 325-326; I'm increasingly concerned that many investigators are using the term "photosynthetic efficiency" for variable fluorescent measurements. F_v/F_m is, as defined, is the quantum yield of PSII fluorescence-it is a reflection of the number of functioning PSII reaction centers. Many other variables are needed to extend that to "photosynthetic efficiency", especially when one also includes all sources of quenching.

Lines 333-336; So, here is an important issue. You have two major photosynthetic players in close proximity; the algal symbionts of the corals and the photosynthetic cyanobacteria of BBD. They have different photosystem/accessory pigment arrangements. So, how did you de-convolute the fluorescence signals from these two entities in such close proximity? Which "color" PAM, red or blue, did you use? This is important because an extended dark period is required to drain the photosystems of electrons in cyanobacteria which still leads to State 2, rather than State 1, conditions which can lead to underestimations of F_m . There is also a well described interaction between cyanobacterial respiratory and photosynthetic electron transport chains because they are found in the same membrane, where electrons from respiration can energize thylakoid membranes. And this is also why an extended period of darkness is required to get more accurate F_m measurements. A red PAM can be more specific for cyanobacteria if set-up correctly, but a blue PAM is more problematic because it will excite both cyanobacterial and eukaryotic algal photosystems. None of this was described and if not considered casts some doubt on the absolute, but probably not the relative, accuracy of the PAM measurements reported.

Line 414; Why did you limit timeframe of literature search from 2006-2023. There is a lot of literature on BBD prior to 2006? If this was done for a better match with Coral Reef Watch data see my next comment.

Line 423; I have said this repeatedly in other reviews so I will just copy and paste; "This reviewer believes the NOAA DHW is a flawed metric. DHW are notoriously variable and based on the premise, from the 1990's, that bleaching occurs universally at 1°C above the mean monthly average for a geographic area. First, that doesn't always hold true. Second, the data used are large scale, coarse, remotely sensed SST which looks at the skin temperature of the oceans (<<10 m). A recent study McClanahan et al. 2019, Nat Clim Change, 9:845-851, showed that DHW in either a multivariate or univariate model predicted only 5% and 9% of the variation between temperature and bleaching". So, we have a what appears to be a truncated literature search based on the availability of a potentially flawed metric of thermal stress. The authors should articulate clearly why they would disagree with this statement, and/or amend/omit this section entirely. Also, McClanahan et al. is not the only paper out there suggesting DHW is flawed.

Lines 430-437; There are no measurements of inorganic nutrients from the Shark River outflow available from Lapointe et al. (2019). There are correlations between outflow and changes in nutrient concentrations and sometimes nutrient stoichiometry that are disconnected spatially and temporally with nutrients actually measured at Looe Key. And again, see comments in Lesser (2021) regarding how few measurements actually exceed commonly accepted threshold, and ratio, values for ecological consequences. From Lesser (2021);

“In fact, calculations for the role of the Everglades in nutrient loading across the Florida Keys suggest that it is minor compared with offshore inputs, such as those from internal waves, on an annual basis (Lamb and Swart 2008, Gibson et al. 2008). Despite a strong relationship between coral cover decline and DIN concentrations that explains 60% of the variance (figure 5a in Lapointe et al. 2019), there have been no significant differences in DIN concentrations since the 1990s and the 2000s for PO₄³⁻ concentrations. All mean values are at or barely greater than the threshold values at Looe Key (table 2 in Lapointe et al. 2019), and the N:P ratios are well below those used by Wiedenmann and colleagues (2013), in most years, to experimentally show increased sensitivity to thermal and light stress, causing coral bleaching.”

Back to Results

Line 90; But Fig. S1 indicates an effect of treatment on quantum yields? Notably, Fig. S1 also reports a significant interaction making those results uninterpretable because treatment effects are now confounded by time. The next step should have been multiple comparison testing on the matrix of treatment and time results to see which factors were important, or not.

Line 98; How is stress defined with respect to quantum yield measurements, especially given there is no indication the authors tried to separate the quantum yields of the cyanobacteria from the Symbiodiniaceae?

Line 100; Quantum yields of PSII fluorescence are not equivalent to photosynthesis. See Warner et al. (2010).

Line 108 and beyond; So, for the metagenetic analysis of 16S rRNA we have samples collected on Days 50 and 73 for healthy tissues, and then samples of BBD infected colonies on day 60. Sampling at different times confounds statistical analysis. Treatment effects are now confounded by time. If you could not sample all treatments/controls on the same day then you should have sampled a subset of all treatment/control samples on different days, and then block for time in your analysis. This is a significant issue that may require new experiments for this portion of the study.

Line 116; “Pseudomonadota” should be italicized. Specifically, here, and throughout the manuscript the taxonomic convention for all Eubacteria/Archaea is that all taxon levels are italicized, from Kingdom to species, per the International Code of Nomenclature of Prokaryotes. Prokaryotic Code (2022 Revision), and The International Committee on Systematics of Prokaryotes (ICSP).

I would also have like to see a more detailed analysis of the cyanobacteria in these BBD experiments given there is some variability reported on the consortium of cyanobacteria found in BBD. Maybe a supplemental table down to genus level?

Lines 150 and beyond; I’ve already discussed above my concerns with the DHW metric. Now, we get to the N:P ratios used for the big picture component. As described the authors only use nitrate for the N calculation where by definition it should include nitrate, nitrite and ammonium. But Bio-Oracle does not provide nitrite or ammonium. The authors should say this as well as any comments on quality control of the data, whether the data are sea surface or benthos and a list of all sites extracted with specific dates and nutrient values for data collection should be provided. No discussion for the quality of this data is provided for the reader to evaluate. The primary reference in Global Ecology and Biogeography is not even provided.

Line 183; How is “nitrogen-limited” defined?

Line 210; Refer to above where reviewer requests list of cyanobacterial taxa, to genus, recovered in their samples.

Lines 241-254; So, members of the genus *Roseofilum*, now described as the primary cyanobacterial component of BBD in Caribbean corals and present in BBD for Pacific corals does apparently have the ability to fix nitrogen (Meyer et al. 2017), but no rates of nitrogen or carbon fixation are available. So, no way of assessing their importance in this regard to the establishment and progression of BBD.

Line 266; Should quote Lesser et al. (2007) here as they were one of the few who recognized the opportunistic nature of not just BBD, but most “coral diseases” based on prior exposure to a number of environmental conditions, including nutrient perturbations.

Figures are fine as are Supplemental files.

Version 1:

Reviewer comments:

Reviewer #1

(Remarks to the Author)

The authors have responded carefully and satisfactorily to all of my comments. The revisions have strengthened the manuscript, and the requested changes have been implemented appropriately. I have no further major concerns.

Reviewer #2

(Remarks to the Author)

The authors have done an excellent job of responding to the comments by both Reviewers. The results is a much more informative paper for the community. I have no additional comments to make.

Breakdown of microbial networks links nutrient stress and reef coral disease

Point-by-point response to Reviewers

[Reviewer comments are shown in blue, responses in black, final reference numbering is shown in the revised manuscript text; newly added references are listed here without numbers for ease of identification.]

Reviewer #1

This is a well-conducted study that advances our understanding of the mechanisms underlying coral disease, specifically the black band disease (BBD). It provides critical insights into how environmental factors, specifically nutrient imbalances, destabilize coral-associated microbiomes and promote disease. It fills a gap in understanding the mechanisms behind black band disease (BBD) outbreaks. The lab experiments are clearly groundbreaking and demonstrated that skewed nitrogen-to-phosphorus (N:P) ratios in seawater significantly contribute to the onset of BBD in corals. The analysis of the coral-associated microbiome from the lab experiments clearly showed disruption of the microbial network, its structure, and connectivity, in the disease tissue. Skewed N:P ratios increase modularity and reduce connections between microbial taxa, leading to fragmented and less resilient microbial communities. The global analysis of BBD outbreaks in relation to temperature and nutrient data strongly supports the conclusions. The worldwide analysis was consistent with the results from the laboratory experiment, revealing 87% of BBD outbreaks occurred in regions with skewed N:P ratios. In comparison, only 13% were linked to prior heat stress. This suggests that nutrient stoichiometry is a stronger determinant of BBD prevalence than thermal stress.

We appreciate the positive comments of the reviewer.

However, I am missing a more thorough discussion and presentation of the bacterial taxa detected in the microbial mats from the experiments. I don't think that the Phylum-level taxonomic resolution is sufficient to claim similarity to the other taxa commonly found in BBD worldwide. I would recommend genus-level taxonomic presentation when possible, or at minimum Order-level resolution

In addition to the phylum-level comparison, we had based our conclusions on the phylogenetic tree of the 16S rRNA amplicon sequences (Suppl Fig. 3) which showed the high degree of similarity between the disease consortia in our experiments and those from environmental samples. However, we fully agree that the taxonomic analysis can be expanded to provide a broader comparative picture of the microbial communities in the BBD lesions from this study. Therefore, we have introduced the following changes in the revised manuscript:

(i) We generated and included a *Taxonomic Alignment* (New **Supplementary Table 4**, included at the end of this document) that highlights the similarity across taxonomic levels between prokaryotes detected in our study and BBD-relevant taxa identified in environmental samples, ranging from the phylum to the genus level. The table focuses on the key functional groups involved in the initiation and progression of BBD: (1) *Cyanobacteriota*, (2) sulfate-reducing bacteria, (3) sulfide-oxidising bacteria, and (4) heterotrophic or opportunistic taxa. In addition, the table includes relevant references to supporting field-based studies. Again, this analysis reveals a high degree of similarity between the taxa identified in our study and those reported from field samples.

(ii) We complemented the high-resolution taxonomic analysis of the *Cyanobacteriota* by a phylogenetic analysis of another key group consistently involved in BBD development, the sulfate-reducing bacteria (*Thermodesulfobacteriota*), because of their role in producing toxic sulfides. The phylogenetic tree (New **Supplementary Figure 3B**, included at the end of this document) provides further evidence of the close similarity of taxa in our study and those isolated from BBD mats in the field.

We have expanded the relevant parts in the results and discussion sections to read:

“Taxonomic alignment (Suppl Table 4) demonstrates a high level of similarity between prokaryotes identified in this study and BBD-associated taxa reported from field samples, spanning key functional groups involved in BBD initiation and progression, namely (1) Cyanobacteriota, (2) sulfate-reducing bacteria, (3) sulfide-oxidising bacteria, and (4) heterotrophic or opportunistic taxa.”

“This is further underpinned by a high-resolution taxonomic and phylogenetic similarity of sulfate-reducing bacteria found in our experiments and established taxa involved in the BBD disease etiology.”

*I would also recommend mentioning that the conclusions are based on one coral species, and testing the effects of nutrient stoichiometry on other coral species beyond *Turbinaria reniformis* would provide broader validation. I think it's possible that other environmental stressors could trigger microbial imbalance in different coral species, resulting in a similar disease presentation. Minor revisions to address confounding factors and expand the discussion would strengthen the paper.*

We have added the following statement to the discussion section:

“Future research should test the effects of nutrient stoichiometry on additional coral species and consider also other environmental stressors that could potentially trigger microbial imbalance causing BBD”.

Reviewer #2:

This an interesting manuscript describing the relationship between nutrient stoichiometry and susceptibility to coral disease and mortality. The data available to our community, but too often overlooked, clearly shows that the majority of “coral diseases” are secondary to some sort of environmental insult (e.g., eutrophication, temperature stress, ocean acidification, etc.). Here, the authors extend their previous studies to examine the effects of deviations in nutrient stoichiometry on coral diseases. The authors use what is a well describe disease, with a primary pathogenic consortium of microbes, to demonstrate the role of nutrient stoichiometry in the initiation and pathogenesis of this disease.

We thank the reviewer for the positive remarks on our work.

My specific concerns can be found below.

***Line 41:** First, While *V. coralliilyticus* appears to be a good example of temperature dependent virulence increasing the prevalence of bacterially mediated coral bleaching, *Vibrio shilonii* and*

V. mediterranei are synonymous, so *V. shilonii* doesn't even exist as *V. mediterranei* has priority from a taxonomic perspective. Second, since 2004, *V. shilonii* has no longer been detectable in either healthy or bleached corals using culture methods, fluorescent in situ hybridization (FISH), or fluorescence microscopy using specific anti-*V. shilonii* antibodies. See Ainsworth et al. 2008, *ISME J*, 2:67-73 which also questions whether *V. shilonii* was ever the cause of bacterial bleaching.

Thank you, we agree. We have removed *V. shilonii* (and associated reference) from the relevant sentence in the manuscript, which now reads:

“Some, such as Stony Coral Tissue Loss Disease (SCTLD), are closely associated with specific bacterial genera while others can be caused by individual, contagious pathogenic species such as Vibrio coralliilyticus”.

Line 43; *History is important. Antonious (1973) and Garrett and Ducklow (1975) were the first to describe BBD in corals.*

We have added the suggested references:

- (i) Antonious A. (1973). *10th Meeting of the Association of Island Marine Laboratories of the Caribbean*, University of Puerto Rico (Mayaguez).
- (ii) Garrett P, Ducklow H (1975) Coral diseases in Bermuda. *Nature* 253:349–350

Line 45; *I see nothing in Sato et al. or in Aronson et al., quoted by Sato, to support the statement that BBD is a major source of coral mortality in the Caribbean. Dinsdale (200) reported a disease prevalence of 1.3% to 4.9% in the Caribbean. Edmunds (1991) reported prevalences as low as 0.2%.*

Thank you for spotting this. The citation of the paper by Sato et al in this sentence was an oversight. We acknowledge that BBD disease prevalence can be quite low in terms of percentage of total coral population. We intended to emphasize that BBD is frequently observed, and its impact on coral reefs is multifactorial and severe (Reviewed by Bruckner 2002 and Sutherland et al 2004). Importantly, BBD has been shown to affect particularly large, slow-growing colonies (Voss and Richardson 2006) which are key for reef framework building (Rützler et al 1983). We have reworded the relevant sentence to reflect more accurately the intended significance in the introduction to our work, considering also the references suggested by the reviewer. This section now reads:

“The BBD prevalence in parts of the Great Barrier Reef and the Caribbean has been shown to temporarily reach up to the order of 5% (Dinsdale, 2002; Muller & Van Woesik, 2011). On other occasions, fewer than 1% of coral colonies on any reef area at any one time were affected by BBD, yet the susceptibility of major frame-building species greatly enhances the impact on coral communities (Sutherland 2004). Furthermore, since the disease occurs repeatedly and affects numerous coral species across the globe, it is an important contributor to reef degradation (Rützler 1983, Voss and Richardson 2006, Bruckner, 2002; Sutherland et al 2004).”

Bruckner, A. W. (2002). *Priorities for effective management of coral diseases* (pp. 57-57). US Department of Commerce, National Oceanic and Atmospheric Administration, National Marine Fisheries Service.

Dinsdale, Elizabeth. (2002). Abundance of black-band disease on corals from one location on the Great Barrier Reef: a comparison with abundance in the Caribbean region. Proc 9th Int Coral Reef Symp, Bali, Indonesia. 2.

Rützler, K., Santavy, D. L., & Antonius, A. (1983). The black band disease of Atlantic reef corals, III: Distribution, ecology, and development.

Sutherland, K. P., Porter, J. W., & Torres, C. (2004). Disease and immunity in Caribbean and Indo-Pacific zooxanthellate corals. *Marine ecology progress series*, 266, 273-302.

Muller, E. M., & Van Woesik, R. (2011). Black-band disease dynamics: prevalence, incidence, and acclimatization to light. *Journal of experimental marine biology and ecology*, 397(1), 52-57.

Line 49; The most common description of this polymicrobial disease is that while the filamentous bacteria are the primary invader, it is the photosynthate from these cyanobacteria and the hypoxic/anoxic conditions created in the mat that fosters the growth of sulfate reducers with their production of hydrogen sulfide, as well as cyanobacterial microcystins, that is the decisive cause of coral mortality.

Thank you for providing a clearer and more concise wording for these elements. We have edited the relevant paragraph, which now reads:

“The disease is characterised by a dark microbial mat that migrates across coral colonies, killing tissue at the front and leaving bare skeleton behind¹⁵. The mat community is dominated by phycoerythrin-rich, gliding, filamentous cyanobacteria that lend the band its characteristic dark pigmentation¹⁷. Photosynthates from the cyanobacteria and the hypo- and anoxic conditions created in the mat foster the growth of sulfate-reducers which produce toxic hydrogen sulfide, which is, together with cyanobacterial microcystins, the decisive cause of coral tissue mortality^{14, 18}.”

Line 57; Borger (2004), reference 29, studied primarily DSS and provided no analysis of temperature versus the prevalence of BBD.

We had included the work by Borger (2004) in this context as it demonstrates that DSS is exacerbated by high temperature and that in turn, corals with DSS were more susceptible to BBD, providing an indirect link between high-temperature and BBD. However, we agree that this is not a direct link and have therefore removed the reference from this context.

Line 59; Sato et al. (2016), a review, does cover the role of nutrients in BBD, but there is no work on nutrients within Muller and van Woesik (2011). Voss et al. (2007) showed no effect of nutrient addition on BBD communities and Voss and Richardson (2006) used profound deviations in nutrient stoichiometry in their experimental treatment over time ending with an N:P of ~58 which you probably could use to make your case. Also, Kuta and Richardson (2002) could/should be added and replace others that are not relevant as suggested above.

Thanks for spotting this mix-up of references. We have removed Muller & van Woesik (2011) and Voss et al. (2007) and have added Kuta and Richardson (2002) instead in this context.

Thank you also for pointing out the stoichiometry aspects of the experimental work of Voss and Richardson (2006). This provides indeed further support of our case, and we have integrated this into the introduction. We have edited the relevant section which now reads:

“Exceptions include studies from the Bahamas and Florida Keys, which provide the concentration of seawater nutrients during the time of the BBD observations^{31, 32}. With reported values of up to ~30, the ratio of total dissolved inorganic nitrogen (N) to soluble phosphorus (P) in sites with BBD was considerably higher compared to locations without BBD (N:P ~22) (Kuta and Richardson 2002). Notably, experimental nutrient enrichment that enhanced BBD progression (Voss and Richardson, 2006) was associated with increases in N:P ratios from ~22 to ~58 under in situ enrichment conditions and from ~20 to ≥62 in tank experiments. In the latter case, the rate of BBD progression increased as the N:P ratio rose, driven by nitrate addition to the seawater (Voss and Richardson, 2006).”

Line 69; *I don't see any nutrient concentration measurements from actual coral reef waters in Rosett et al. (2017), reference 33, as suggested by the text. Use the original, primary, references where those values came from.*

We have replaced the citation Rossett et al 2017 in this context and used instead the Kuta and Richardson (2002) reference suggested by the reviewer (see response to previous comment). This facilitates a more direct comparison of BBD and non-BBD sites.

Line 71; *Lapointe et al. (2019) does discuss, extensively, the relationship between altered nutrient stoichiometry and coral bleaching, but not for coral growth, only for macroalgae. Also, Lesser (2021) discusses at length a number of issues and misinterpretations of how often both eutrophication, and deviations in nutrient stoichiometry, actually occurred in the Florida Keys during the 30 y time period reported by Lapointe et al. (2019).*

We have clarified the link between the cited references and the effects of skewed N:P ratios by including them at the relevant position within the sentence. We have also included the reference Lesser (2021) to provide a broader perspective. This section reads now:

“Such skewed dissolved inorganic N:P ratios can have serious detrimental effects on symbiotic reef corals³⁶ (Lesser et al., 2021), ranging from increased bleaching susceptibility^{33, 34, 38} to reduced growth^{35, 37}.”

Lines 295-296; *Please report the exact number of genets and ramets used in the experiment, and how they were randomized in the experiment. Was genet considered as a factor in the experiment? If not, why not?*

We used 18 colonies, originally derived from a single mother colony to reduce genetic variance as a confounding factor in our model system. Accordingly, genets were not considered as factor. Using representative models with no or very low genetic variability among replicates is a standard approach in mechanistic, biological studies (e.g. *E. coli* and other bacterial model strains, Yeast, HeLa and other immortalised cell lines, *Aiptasia*, *Symbiodinium* and other algal cultures). Still, while genetically identical,

each experimental colony was grown from a small replicate and has therefore an individual life history, in other words, the replicate colonies are not just small fragments of a broken up large colony. The sentence reads now:

*“A single mother colony of *T. reniformis* was fragmented and the fragments were grown into replicate colonies for >6 months. The replicate coral colonies (n = 18) were then cultured under the three different seawater N:P stoichiometry systems for >7 weeks.”*

In response to a comment from reviewer 1, we have also added the following statement in to the discussion:

“Future research should test the effects of nutrient stoichiometry on additional coral species and consider also other environmental stressors that could potentially trigger microbial imbalance causing BBD”.

Line 320; *What is the ecological context for using irradiances of 120 $\mu\text{mol quanta m}^{-2} \text{s}^{-1}$? At a depth of 15 m irradiances can vary from ~500-1000 $\mu\text{mol quanta m}^{-2} \text{s}^{-1}$ on many Caribbean and Pacific coral reefs (see Lesser 2024). An irradiance of 120 $\mu\text{mol quanta m}^{-2} \text{s}^{-1}$? is therefore occurring at deeper depths where it has been noted that BBD actually declines in prevalence.*

These considerations are correct. However, while being shallow water corals, our model coral *Turbinaria reniformis*, as well as the closely related species *T. mesenterina*, are frequently found in turbid waters with lower light levels (Veron, 2000). Field measurements have reported ~120 $\mu\text{mol m}^{-2} \text{s}^{-1}$ in the proximity of *T. mesenterina* at 4m depth (Juhi et al. 2021). This paper also found that corals from this habitat retained close-to-natural symbiont densities in an aquarium setting when kept at these light levels on a 12h light/dark cycle. Furthermore, Hoogenboom et al (2009) found that optimal growth irradiances are situated towards the lower end of the habitable light intensity range of *Turbinaria*, peaking at ~300 $\mu\text{mol m}^{-2} \text{s}^{-1}$. This is in line with care instructions of ornamental traders who recommend light regimes in the order of 150 $\mu\text{mol m}^{-2} \text{s}^{-1}$ for *Turbinaria* (https://coralaxy.de/en/products/turbinaria-peltata-green?srsId=AfmBOoq5fZ9FFxgjkIvTCubC7bMnKeWnkOLFp1qi_zMRTsoo8tgvjULv) and our own experience in growing of *Turbinaria* in the laboratory for many years. We note that our system is operated on a 12h light/dark cycle, providing constant light intensities with only short ramping times at the beginning and end of the light period. Accordingly, corals receive a higher cumulative photon dose compared to corals in the field in deeper habitats where maximal levels of 120 $\mu\text{mol quanta m}^{-2} \text{s}^{-1}$ may be reached around midday.

Since our experiments were designed to test the impact of altered nutrient environments, we aimed to keep other parameters such as light in the ambient range for our model coral. We have added the following information to the method section for clarification:

*“Corals were cultured at $26\pm 1^\circ\text{C}$, salinity 33 ± 0.5 , and a photon flux of $\sim 120 \mu\text{mol m}^{-2} \text{s}^{-1}$ under a 12:12h light/dark cycle^{33, 34}. The light regime was selected based on light levels in the natural habitats in 4m depth of the closely related species *T. mesenterina* (Juhi et al. 2021) and on evidence that optimal growth irradiances for this species occur toward the lower end of its habitable light intensity range, peaking at approximately $300 \mu\text{mol m}^{-2} \text{s}^{-1}$ (Hoogenboom et al. 2009). The lighting, heating,*

water movement, filtration and macronutrient concentration parameters for the experimental system have been described previously^{34, 78}.

Hoogenboom, M. O., Connolly, S. R., & Anthony, K. R. (2009). Effects of photoacclimation on the light niche of corals: a process-based approach. *Marine Biology*, 156(12), 2493-2503.

Juhi, Z. S., Mubin, N. A. A. A., Jonik, M. G. G., Salleh, S., & Mohammad, M. (2021). Impact of short-term light variability on the photobiology of turbid water corals. *Journal of Sea Research*, 175, 102088.

Veron, J.E.N. (2000) *Corals of the World*. Vol. 1-3, Australian Institute of Marine Science, Townsville, 1382.

Lines 325-326: *I'm increasingly concerned that many investigators are using the term "photosynthetic efficiency" for variable fluorescent measurements. Fv/Fm is, as defined, is the quantum yield of PSII fluorescence-it is a reflection of the number of functioning PSII reaction centers. Many other variables are needed to extend that to "photosynthetic efficiency", especially when one also includes all sources of quenching.*

We have edited the text and changed the relevant wording to "maximum PSII quantum yield (Fv/Fm)".

Lines 333-336: *So, here is an important issue. You have two major photosynthetic players in close proximity; the algal symbionts of the corals and the photosynthetic cyanobacteria of BBD. They have different photosystem/accessory pigment arrangements. So, how did you deconvolute the fluorescence signals from these two entities in such close proximity? Which "color" PAM, red or blue, did you use? This is important because an extended dark period is required to drain the photosystems of electrons in cyanobacteria which still leads to State 2, rather than State 1, conditions which can lead to underestimations of Fm. There is also a well described interaction between cyanobacterial respiratory and photosynthetic electron transport chains because they are found in the same membrane, where electrons from respiration can energize thylakoid membranes. And this is also why an extended period of darkness is required to get more accurate Fm measurements. A red PAM can be more specific for cyanobacteria if set-up correctly, but a blue PAM is more problematic because it will excite both cyanobacterial and eukaryotic algal photosystems. None of this was described and if not considered casts some doubt on the absolute, but probably not the relative, accuracy of the PAM measurements reported.*

These are fair concerns. However, when taking the Fv/Fm measurements of fully (>10h) dark acclimated corals, we had ensured that the measured area on the coral was at least 3 mm away from any microbial mats in the surrounding. Below, we provide the results of additional control measurement that underpin the validity of this approach.

1) A control measurement demonstrates that the fluorescence of the microbial mats is only detectable if there is a direct overlap of the measuring spot of the PAM probe and the mat (Rebuttal Figure R1). Measuring coral Fv/Fm even in proximity of mats will not pick up fluorescence signals from them. Furthermore, the fluorescence intensity recorded by the PAM is less than 50% of that of the symbionts in the coral tissue

(Rebuttal Figure R2B) in a relevant measuring distance of ≤ 5 mm, further reducing the risk that microbial fluorescence signals are measured during coral monitoring.

2) The fluorescence signals of the model coral and the microbial mats are distinct and spatially well separated as demonstrated by fluorescence microscopic images (Rebuttal Figure R2). Fluorescence emission spectra of coral tissue are dominated by the green fluorescence of host pigments together with a red fluorescence signal from the zooxanthellae chlorophyll (Rebuttal Figure R2). In contrast, the microbial mats are dominated by the emission of orange and red fluorescence that can be attributed to cyanobacterial phycobiliproteins and bacterial chlorophylls. The host tissue reveals only a minor contribution of these spectral features to the fluorescence signature in this wavelength window, indicating that the cyanobacteria present in the coral tissue will not make a significant contribution to the recorded Fv/Fm values.

3) A comparable reduction in Fv/Fm under similar conditions has been recorded previously in the other species in the absence of microbial mats (Rosset et al., 2017; Buckingham et al., 2023).

We have adjusted the method section to provide additional information to improve clarity. This paragraph reads now:

“Photosystem II (PSII) maximum quantum yield (Fv/Fm) was measured after ≥ 10 h of dark acclimation using a submersible pulse-amplitude modulated fluorometer (Diving-PAM) (Walz) as proxy of the overall health of Symbiodiniaceae endosymbionts⁴². Declining Fv/Fm values were considered as indicative of stress experienced by the symbionts (Rosset et al 2017; Buckingham et al, 2023). This PAM model excites fluorescence by pulse modulated red light from a light-emitting-diode (LED, 655 nm). When recording Fv/Fm measurements of the coral tissue, we ensured that the measuring area of the PAM was located at least 3 mm away from microbial mats in the proximity to prevent a spectral contamination by fluorescence signals from the microbes.”

Figure R1: Fluorescence (F) recorded with the Diving PAM (Walz) of a white ceramic tile placed next to a coral fragment covered by a BBD microbial mat. The upper graph shows the corresponding F values. The lower image shows the measured areas on the tile, the coral and the interface area.

Figure R2: Fluorescence properties of the experimental coral (*cor*) *Turbinaria reniformis* partially covered with a microbial mat (*mat*). (A) Daylight image taken under a fluorescence microscope (Leica MZ10) of a coral sample with a microbial mat covering the right part of the specimen. A lesion (*les*) revealing bare skeleton is visible in the centre. (B) Fluorescence (F) recorded with the Diving PAM at defined distances from the coral and mat areas indicates a weaker fluorescence intensity of the mat areas. (C) Blue light (450 nm)-excited fluorescence imaged with a long-pass filter reveals green fluorescence of coral tissue, prominently localised in the polyps. (D) Fluorescence emission spectra of coral tissue and mat areas excited with blue light (450nm) show dominant fluorescence of green fluorescent host pigments (GFP) and zooxanthellae chlorophyll (zCHL) in the coral tissue area. Microbial mats are not returning strong signals under these measurement settings. (E) Green light (530 nm)-excited fluorescence imaged with a TRITC filter localises orange-red fluorescence of phycobilin proteins in the mat-forming microbes. (F) Fluorescence emission spectra of coral tissue and mat areas excited with green (520nm) light. The mat area shows fluorescence of cyanobacterial phycobilin proteins (phycoerythrin, PE; phycocyanin/allophycocyanin, PC/APC) and bacterial chlorophyll (bCHL). Coral areas are dominated by red fluorescence emitted by zooxanthellae chlorophyll (zCHL). Error bars show standard deviations of readings from different sample areas (n=4).

Line 414: *Why did you limit timeframe of literature search from 2006-2023. There is a lot of literature on BBD prior to 2006? If this was done for a better match with Coral Reef Watch data see my next comment.*

We had used the key publication of Voss and Richardson (2006) as a starting point for the literature search as it provided relevant data of BBD-occurrence as well as nutrients levels in the seawater. However, we have now extended our analysis back to the year 2000. This is the limit set by the availability of nutrient data from Bio-Oracle required for the study. Extending the period to 24 years allowed the inclusion of further 18 papers in the analysis, reaching a total of 141 studies (230 BBD incidence sites). The new data points provided additional evidence for the occurrence of BBD in sites with skewed N:P ratios without affecting on the overall results and conclusions. The updated analysis is shown in:

(1) Revised Figure 4

(2) Updated Supplementary Table 1

(3) The relevant paragraph in the method section now reads:

“A literature search was conducted using Web of Science. A search for research papers published in the 24-year period between 2000 and 2023 including “black band disease” in either the title or abstract was conducted. The cut-off in the year 2000 was introduced based on the availability of corresponding nutrient data from the Bio-Oracle data base. Each paper was downloaded and reviewed individually....”

Line 423: *I have said this repeatedly in other reviews so I will just copy and paste; “This reviewer believes the NOAA DHW is a flawed metric. DHW are notoriously variable and based on the premise, from the 1990’s, that bleaching occurs universally at 1°C above the mean monthly average for a geographic area. First, that doesn’t always hold true. Second, the data used are large scale, coarse, remotely sensed SST which looks at the skin temperature of the oceans (<<10 m). A recent study McClanahan et al. 2019, Nat Clim Change, 9:845-851, showed that DHW in either a multivariate or univariate model predicted only 5% and 9% of the variation between temperature and bleaching”. So, we have a what appears to be a truncated literature search based on the availability of a potentially flawed metric of thermal stress. The authors should articulate clearly why they would disagree with this statement, and/or amend/omit this section entirely. Also, McClanahan et al. is not the only paper out there suggesting DHW is flawed.*

We are aware of the McClanahan et al. paper and fully agree with their conclusions and the reviewers’ concerns. In fact, the variability in nutrient stoichiometry across sites identified in this study may potentially influence bleaching tolerance (sensu Wiedenmann et al. 2013), providing an alternative pathway to explore the reasons for the observed geographical variation and reduced predictability of bleaching events found by McClanahan and others. **Importantly, in the present study, DHW was used solely as an established metric to quantify environmental heating, without attempting to directly link this metric to bleaching occurrence, or even to stress level experienced by the corals.**

At present, the NOAA Coral Reef Watch DHW and related products represent unique datasets facilitating comparisons of water temperature anomalies across global reef environments on decadal timescales, yet with a high temporal resolution. Consequently, while DHW may not be ideal for predicting bleaching per se, it provides -within the technical constraints noted by the reviewer- a robust indicator of whether BBD incidence tends to increase following periods of elevated environmental heat levels. The fact that we observed such a relationship in only very few cases suggests that increased sea water temperature is not the primary driver of BBD development, particularly given the strong statistical support for skewed nutrient ratios as a key contributing factor. Please see also our response above regarding the expanded literature review. The individual data points are provided in the newly introduced Supplementary Table 5 (included at the end of this document).

Accordingly, we have modified the relevant paragraph in the method section which now reads:

“The NOAA Coral Reef Watch (CRW) daily 5-km satellite coral bleaching Degree Heating Week (DHW) product (Liu et al, 2014) (version 3.1, released August 1, 2018) was used to reconstruct the thermal history of reported BBD events. The Degree Heating Weeks (DHW) data show the accumulated heating levels for a specific location with 5km resolution. We used DHW data for the 365 days prior to a BBD event as metric for anomalously high temperatures experienced by each BBD incidence point (Suppl Table 5).”

Liu, G.; et al, Reef-Scale Thermal Stress Monitoring of Coral Ecosystems: New 5-km Global Products from NOAA Coral Reef Watch. Remote Sens. 2014, 6, 11579-11606.

Lines 430-437: There are no measurements of inorganic nutrients from the Shark River outflow available from Lapointe et al. (2019). There are correlations between outflow and changes in nutrient concentrations and sometimes nutrient stoichiometry that are disconnected spatially and temporally with nutrients actually measured at Looe Key. And again, see comments in Lesser (2021) regarding how few measurements actually exceed commonly accepted threshold, and ratio, values for ecological consequences. From Lesser (2021): “In fact, calculations for the role of the Everglades in nutrient loading across the Florida Keys suggest that it is minor compared with offshore inputs, such as those from internal waves, on an annual basis (Lamb and Swart 2008, Gibson et al. 2008). Despite a strong relationship between coral cover decline and DIN concentrations that explains 60% of the variance (figure 5a in Lapointe et al. 2019), there have been no significant differences in DIN concentrations since the 1990s and the 2000s for PO₄³⁻ concentrations. All mean values are at or barely greater than the threshold values at Looe Key (table 2 in Lapointe et al. 2019), and the N:P ratios are well below those used by Wiedenmann and colleagues (2013), in most years, to experimentally show increased sensitivity to thermal and light stress, causing coral bleaching.”

We agree that the overall nutrient budget of the Florida Keys region cannot be directly explained by land-derived nutrients alone. We have therefore added further references, including those suggested by the reviewer, to provide a broader perspective. Importantly, the key observation that elevated N:P ratios can be linked to an increased BBD prevalence and the derived conclusions are not altered by these considerations. The relevant paragraph in the method section reads now:

“We note in this context that the Everglades run-off contributes only a part of the nutrients that are delivered from the wider region to these reefs through oceanographic processes (Leichter et al. 1996; Boyer & Jones 2002; Lamb and Swart 2008, Gibson et al 2008). Hence, it is not possible to directly link the unusually high N:P ratio of the region (Jung et al. 2025, Boyer & Jones 2002) to the nutrient enrichment through coastal processes alone (Lesser 2021).”

The relevant results section reads:

“Across the 24-year study interval, 36 incidences of BBD were reported off the southeast coast of Florida and the Florida Keys. Of the 15 BBD incidences for which both location and time of identification were reported, prior heat stress exposure (DHW=5) was detected in only two cases (Suppl Fig 5A). Instead, the whole affected region is characterised by N:P ratios of 30 to 560, with land-derived nutrients primarily delivered through the Everglades including the Shark River Slough⁴⁶ (Suppl Fig 5B). The nutrient introduction into coastal waters increases with the amount of runoff and riverine discharge^{47, 48}. The daily discharge volume, and thereby nutrient influx, from the Shark River Slough was above the long-term average for an average of 47 days in the years with reported BBD outbreaks (Suppl Fig 5C).”

Boyer, J. N., & Jones, R. D. (2002). A view from the bridge: external and internal forces affecting the ambient water quality of the Florida Keys National Marine Sanctuary (FKNMS). The Everglades, Florida Bay, and Coral Reefs of the Florida Keys: An Ecosystem Sourcebook. CRC Press, Boca Raton, FL, 609-628.

Gibson, P. J., Boyer, J. N., & Smith, N. P. (2008). Nutrient mass flux between florida bay and the florida keys national marine sanctuary. *Estuaries and coasts*, 31(1), 21-32.

Jung, J., Ardisana, R., Vermeij, M. J., & Murphy, E. L. (2025). Physiological consequences of nitrogen enrichment for corals in the Caribbean. *Royal Society Open Science*, 12(6), 250208.

Lamb, K., & Swart, P. K. (2008). The carbon and nitrogen isotopic values of particulate organic material from the Florida Keys: a temporal and spatial study. *Coral Reefs*, 27(2), 351-362.

Leichter, J. J., Wing, S. R., Miller, S. L., & Denny, M. W. (1996). Pulsed delivery of subthermocline water to Conch Reef (Florida Keys) by internal tidal bores. *Limnology and Oceanography*, 41(7), 1490-1501.

Lesser, M. P. (2021). Eutrophication on coral reefs: what is the evidence for phase shifts, nutrient limitation and coral bleaching. *BioScience*, 71(12), 1216-1233.

Line 90; But Fig. S1 indicates an effect of treatment on quantum yields? Notably, Fig. S1 also reports a significant interaction making those results uninterpretable because treatment effects are now confounded by time. The next step should have been multiple comparison testing on the matrix of treatment and time results to see which factors were important, or not.

Thank you for flagging. We have adjusted the wording of this section (starting at former Line 90) to make it clearer which conditions are referred to. The relevant sentences read now:

“Coral kept under control conditions (High Nitrate : High Phosphate; HN:HP, N:P~8) remained healthy throughout the duration of the 73-day-experiment, with symbionts exhibiting no change in their maximum PSII quantum yield (Fig 1; Suppl Fig 1).....”

“.....In contrast, in the HN:LP treatment, decreasing maximum PSII quantum yield of symbiont photosynthesis was observed (Suppl Fig 1), and disease lesions were present within 10 days (Fig 1E).”

We also see that the choice of wording in the caption of Supplementary Figure 1 was misleading. We have conducted a segmented regression analysis of the HN:LP conditions for clarification which identifies a significant breakpoint at day 16.7 +/- 2.3 SE, transitioning from a rapid decline (slope = -0.0079, $p < 0.001$) to a stable state. No significant declines or breakpoints were identified in Fv/Fm time course of the Control and the LNHP groups. Importantly, the start of sampling at day 50 (black arrow) took place after the corals were fully acclimatised to the respective nutrient conditions. Accordingly, the time-dependent initial drop in Fv/Fm that was observed in the HN:LP treatment is not relevant at the time of sampling. For clarification, we have rewritten the figure legend and introduced explanatory elements in the graph as shown below.

“**Supplementary Figure 1: Effect of seawater N:P stoichiometry on maximum quantum yield of Photosystem II (Fv/Fm).** Mean \pm standard deviation of Fv/Fm values are shown for *T. reniformis* under Control, HN:LP and LN:HP conditions ($n = 6$ replicate colonies per nutrient condition). Type III ANOVA with Satterthwaite’s method indicated significant main effect of nutrient treatment ($F(2, 75.271) = 20.615, p < 0.001$) when comparing HN:LP conditions with either Control or LN:HP conditions. No significant differences were detected between Control or LN:HP conditions. Segmented regression analysis of the HN:LP conditions identified a significant breakpoint at day 16.7 +/- 2.3 SE (red arrow), transitioning from a rapid decline (slope = -0.0079, $p < 0.001$) to a stable state. No significant declines or breakpoints were identified in Fv/Fm time courses of the Control and the LNHP groups. The start of sampling at day 50 (black arrow) took place after the corals were fully acclimatised to the respective nutrient conditions.”

Line 108 and beyond: So, for the metagenetic analysis of 16S rRNA we have samples collected on Days 50 and 73 for healthy tissues, and then samples of BBD infected colonies on day 60. Sampling at different times confounds statistical analysis. Treatment effects are now confounded by time. If you could not sample all treatments/controls on the same day then you should have sampled a subset of all treatment/control samples on different days, and then block for time in your analysis. This is a significant issue that may require new experiments for this portion of the study.

We sampled subsets of n=3 replicate coral colonies from HN:HP (control), LN:HP and HN:LP on both, day 50 and day 73. Importantly, sampling involved separate sets of n=3 replicate colonies for each timepoint and nutrient condition, we did not sample the same colonies repeatedly on days 50 and 73. We analysed if there was a statistically significant difference in beta-diversity among communities between these two timepoints. The results of this analysis were included within the statistical analysis shown in Supplementary Table 2 in the original manuscript (included also here for information).

Supplementary Table 2

...

Analysis	Statistical test	Variable	DF	Sum of sq	R2	F	P value
Comparison of beta-diversity between communities associated with whole coral samples by nutrient conditions and time.							
Homogeneity of variance	betadisper	Nutrient condition	2	0.000953		0.078	0.925
		Residual	15	0.091			
Community structure	PERMANOVA	Nutrient condition	2	1.699	0.261	2.653	0.001
		Residual	15	4.805	0.739		
Combined effects	Two-way PERMANOVA	Nutrient condition	2	1.699	0.261	2.691	0.001
		Timepoint	1	0.330	0.051	1.046	0.372
		Nutrient condition :Timepoint	2	0.685	0.105	1.085	0.340
		Residual	12	3.789	0.582		

...

This analysis shows that the main effect of nutrient stoichiometry significantly altered microbial community composition, with no significant effect of time (“timepoint”) and no significant interaction between time (“timepoint”) and nutrient condition, demonstrating temporal stability in the microbial communities across the sampling period.

However, we acknowledge that this format is not very accessible to the reader. To improve clarity, we also ran additional PERMANOVA analyses which demonstrate that there is no difference in the microbial community structure between sampling days 50 and 73 for each nutrient condition and have included this information (below) in the new version of Supplementary Table 2:

...

Sampling Day	Sampling by nutrient condition		
	HN:HP (Control)	LN:HP	HN:LP
50	n=3	n=3	n=3
73	n=3	n=3	n=3
Statistical analysis of beta diversity between time points.	No change in microbial community composition over time (PERMANOVA: $p=0.4$) and dispersion remained consistent between timepoints ($p=0.6$).	No change in microbial community composition over time (PERMANOVA: $p=0.4$) and dispersion remained consistent between timepoints ($p=0.3$).	No change in microbial community composition over time (PERMANOVA: $p=0.6$) and dispersion remained consistent between timepoints ($p=0.9$).

...

On day 60, we sampled microbial communities of the BBD microbial lesions / mats. In this case, we were primarily interested in whether there are significant differences between the BBD mats that formed under the different nutrient treatments (HN:LP vs LN:HP). We used the opportunity to collect also additional samples of the remaining visually healthy tissue from the same colonies. We have amended the relevant paragraph in the method section (Aquarium experiments) to read:

“Microbial lesion samples (n = 8) were collected on day 60 from HN:LP and LN:HP conditions. For comparison, samples from the remaining healthy tissue of the same colonies were also collected on day 60 from LN:HP (n=1) and HN:LP (n=3). Replicate seawater samples were collected at each sampling timepoint (days 50, 60 and 73; n = 12).”

When we compared BBD mat samples taken on day 60 to the coral samples taken on days 50 and 73, our PERMANOVA analysis shows no significant differences ($p>0.01$) in microbial community composition ($p>0.01$) and no significant difference in dispersion between timepoints ($p = 0.06$). This analysis is included in Supplementary Table 2 and also underpins the beta-diversity analysis of microbial communities presented as NMDS ordination plot of Bray-Curtis distances in Figure 2F (included below for information). The samples of the remaining healthy tissue collected on day 60 were also included in this analysis.

Figure 2F shows that all data points from each nutrient condition (healthy tissue from different timepoints (50/60/73) and disease lesions / mats) form statistically well-supported clusters that are clearly distinct from each other and from the seawater samples. Accordingly, sampling at different time points does not represent a confounding factor in this context.

Fig. 2 Alpha and beta diversity of prokaryotic communities associated with *Turbinaria reniformis* and seawater under control, LN:HP and HN:LP conditions...... F
 NMDS ordination of Bray-Curtis distances showing beta-diversity of rarefied relative abundance ASV counts. Each data point represents a replicate coral colony (spheres), seawater sample (triangles) or disease lesion (squares). Statistical comparisons are detailed in Supplementary Table 2.

Line 98; How is stress defined with respect to quantum yield measurements, especially given there is no indication the authors tried to separate the quantum yields of the cyanobacteria from the Symbiodiniaceae?

Please refer to our response to the reviewer comment regarding ms Lines 333-336 above.

Line 100; Quantum yields of PSII fluorescence are not equivalent to photosynthesis. See Warner et al. (2010).

We have edited the wording in relevant section which now reads:

“In contrast, in HN:LP treatment, decreasing maximum PSII quantum yields were observed...”

Line 116; “Pseudomonadota” should be italicized. Specifically, here, and throughout the manuscript the taxonomic convention for all Eubacteria/Archaea is that all taxon levels are italicized, from Kingdom to species, per the International Code of Nomenclature of Prokaryotes. Prokaryotic Code (2022 Revision), and The International Committee on

Systematics of Prokaryotes (ICSP). I would also have like to see a more detailed analysis of the cyanobacteria in these BBD experiments given there is some variability reported on the consortium of cyanobacteria found in BBD. Maybe a supplemental table down to genus level?

Thank you for pointing out this omission. We have now set all bacterial taxa in the text in italics.

A complete list of the cyanobacteria present in the BBD is provided in the phylogenetic tree in Suppl. Fig. 3A to the best taxonomic resolution available. We also generated and included a *Taxonomic Alignment* (**Supplementary Table 4**, included at the end of this document) that illustrates the taxonomic levels at which prokaryotes detected in our study show similarity to BBD-relevant taxa identified in environmental samples, ranging from the phylum to the genus level. The table focuses on key functional groups involved in the initiation and progression of BBD: (1) *Cyanobacteriota*, (2) sulfate-reducing bacteria, (3) sulfide-oxidising bacteria, and (4) heterotrophic or opportunistic taxa. In addition, the table includes relevant references to supporting field-based studies. Overall, this analysis reveals a high degree of taxonomic similarity among representative members of these key functional groups. Please compare also response to Reviewer 1 for further details.

Lines 150 and beyond: I've already discussed above my concerns with the DHW metric. Now, we get to the N:P ratios used for the big picture component. As described the authors only use nitrate for the N calculation where by definition it should include nitrate, nitrite and ammonium. But Bio-Oracle does not provide nitrite or ammonium. The authors should say this as well as any comments on quality control of the data, whether the data are sea surface or benthos and a list of all sites extracted with specific dates and nutrient values for data collection should be provided. No discussion for the quality of this data is provided for the reader to evaluate. The primary reference in Global Ecology and Biogeography is not even provided.

We agree that a complete dissolved inorganic nitrogen (DIN) budget would ideally include nitrite and ammonium, too. However, data availability is limited and Bio-Oracle v3.0 is currently the best available global dataset for dissolved inorganic nutrients (NO₃, PO₄) (Tyberghein et al., 2012). Therefore, we used the cross-validated Bio-Oracle v3.0 surface nitrate layer as a conservative proxy for dissolved inorganic nitrogen at a global scale. These data are derived from the Global Ocean Biogeochemistry Analysis and Forecast, which is a re-analysis from the Copernicus Marine Environment Monitoring Service. Bio-Oracle v3.0 incorporates ship-based nutrient measurements from oceanographic surveys to create the global product. In approximation, nitrite mostly contributes in the order of 10-14% to combined NO₃/NO₂ values (Wiedenmann et al., 2013, Voss and Richardson 2006) and does therefore not represent a major part of the DIN complements. Furthermore, nitrate is often found in greater concentrations than ammonium in reef waters (Lesser et al. 2021 and references therein), hence, the nitrate data set can serve as a solid indicator of the DIN environment. Still, the Bio-Oracle v3.0 data are likely underestimating the concentrations of total dissolved inorganic nitrogen. Importantly, phosphate measurements are not affected by different measuring and reporting methods. **Therefore, the key message that elevated N:P ratios can be linked with increased BBD prevalence remains valid.** However, we acknowledge that the effective N:P

ratios might be higher in certain cases than suggest by the nitrate-based data. In the context of the reported orders of magnitude, this difference does not appear substantial, though. Specifically, our Bio-Oracle–derived datasets yielded N:P ratios of approximately 30 to 560 for the Florida Keys region over the period 2000–2018, while a dedicated nutrient survey conducted in the same region during 1995–1998 reported median total DIN:P ratios ranging from 32 to 160 (Boyers and Jones, 2002; Fig. 17).

We have included additional information for clarification in the method section along with the primary references which reads now:

“Next, global N:P seawater stoichiometry was calculated from nitrate and phosphate surface seawater (0 to 0.49 meters depth) concentrations obtained from Bio-ORACLE v3.0 (Tyberghein et al., 2012; Assis et al., 2002) using QGIS and the prevailing N:P conditions were extracted for each BBD incidence point. Bio-Oracle does not provide ammonium and nitrite values, which contribute to the total amount of dissolved inorganic nitrogen (DIN). Accordingly, the biological relevant N:P ratios reported here are based on nitrate data only and may be lower compared to the total DIN based ratios experienced by the corals. In the context of the reported orders of magnitude, this difference does not appear substantial. Specifically, our Bio-Oracle–derived datasets yielded N:P ratios of approximately 30 to 560 for the Florida Keys region over the period 2000–2018, while a dedicated nutrient survey conducted in the same region during 1995–1998 reported median total DIN:P ratios ranging from 32 to 160 (Boyers and Jones, 2002; Fig. 17).”

Additionally, we have introduced a new **Supplementary Table 5** (included at the end of this document) that compiles the nitrate and phosphate data retrieved from Bio-Oracle and the derived N:P ratio for each BBD incidence point used in this study, along with the related temperature data from the NOAA Coral Watch data base.

Tyberghein, L., Verbruggen, H., Pauly, K., Troupin, C., Mineur, F., & De Clerck, O. (2012). Bio-ORACLE: a global environmental dataset for marine species distribution modelling. *Global ecology and biogeography*, 21(2), 272-281.

Boyer, J. N., & Jones, R. D. (2002). A view from the bridge: external and internal forces affecting the ambient water quality of the Florida Keys National Marine Sanctuary (FKNMS). *The Everglades, Florida Bay, and Coral Reefs of the Florida Keys: An Ecosystem Sourcebook*. CRC Press, Boca Raton, FL, 609-628.

Assis J, et al. Bio-ORACLE v3.0. Pushing marine data layers to the CMIP6 Earth System Models of climate change research. *Global Ecology and Biogeography* **33**, (2024)

Line 183: How is “nitrogen-limited” defined?

Waterbodies containing the global coral reef belt have been confirmed to be mostly N-limited (Browning & Moore, 2023). In the present manuscript, we use a conservative approach and consider reef waters to be N-limited if the dissolved inorganic N:P ratio is smaller than the canonical Redfield N:P ratio of 16, specifically, when it falls in or below the range of ratios between ~4 and ~7, previously reported for different reef environments (Smith et al., 1981; Crossland et al., 1984; Furnas et al., 1995). We have clarified this in the method section by including the following statement:

“Here, we consider reef waters to be N-limited if the dissolved inorganic N:P ratio is smaller than the canonical Redfield N:P ratio of 16 (Redfield, 1958), specifically it falls in or below the range of ratios between ~4 and ~7, previously reported for different reef environments (Smith et al., 1981; Crossland et al., 1984; Furnas et al., 1995). Ratios between ~4 and ~7 were set as a boundary range to identify conditions with deviating stoichiometry (Rosset et al., 2017).”

Browning, T. J., & Moore, C. M. (2023). Global analysis of ocean phytoplankton nutrient limitation reveals high prevalence of co-limitation. *Nature Communications*, 14(1), 5014.

Crossland, C., Hatcher, B., Atkinson, M., Smith, S., 1984. Dissolved nutrients of a high-latitude coral reef, Houtman Abrolhos Islands, Western Australia. *Mar. Ecol. Prog. Ser.* 14:159–163.

Furnas, M., Mitchell, A., Skuza, M., 1995. Nitrogen and Phosphorus Budgets for the Central Great Barrier Reef Shelf. Great Barrier Reef Marine Park Authority.

Redfield, A.C., 1958. The biological control of chemical factors in the environment. *Am. Sci.* 46, 205–221.

Smith, S.V., Kimmerer, W.J., Laws, E.A., Brock, R.E., Walsh, T.E.D.W., 1981. Kaneohe Bay sewage diversion experiment: perspectives on ecosystem responses to nutritional perturbation I. *Pac. Sci.* 35, 279–395.

***Line 210:** Refer to above where reviewer requests list of cyanobacterial taxa, to genus, recovered in their samples.*

Done. Please see our related responses above and the expanded supplementary material (Supplementary Table 4, included at the end of this document).

***Lines 241-254:** So, members of the genus *Roseofilum*, now described as the primary cyanobacterial component of BBD in Caribbean corals and present in BBD for Pacific corals does apparently have the ability to fix nitrogen (Meyer et al. 2017), but no rates of nitrogen or carbon fixation are available. So, no way of assessing their importance in this regard to the establishment and progression of BBD.*

This is a good thought. We have integrated information on the potential of different cyanobacterial taxa to fix nitrogen in the newly introduced *Taxonomic Alignment* (Supplementary Table 4, included at the end of this document) and found that indeed a considerable number of taxa detected in our study are potentially capable of fixing nitrogen. However, since the reviewer comment still applies, we have added the following statement to the corresponding discussion item:

“Future research should test the rates of nitrogen fixation (Lesser et al., 2007) in different nutrient environments to provide a quantitative framework for this potential mechanism.”

Lesser, M. P., Falcón, L. I., Rodríguez-Román, A., Enríquez, S., Hoegh-Guldberg, O., & Iglesias-Prieto, R. (2007). Nitrogen fixation by symbiotic cyanobacteria provides a

source of nitrogen for the scleractinian coral *Montastraea cavernosa*. *Marine Ecology Progress Series*, 346, 143-152.

***Line 266;** Should quote Lesser et al. (2007) here as they were one of the few who recognized the opportunistic nature of not just BBD, but most “coral diseases” based on prior exposure to a number of environmental conditions, including nutrient perturbations.*

We have added the suggested reference:

Lesser, M. P., Bythell, J. C., Gates, R. D., Johnstone, R. W., & Hoegh-Guldberg, O. (2007). Are infectious diseases really killing corals? Alternative interpretations of the experimental and ecological data. *Journal of experimental marine biology and ecology*, 346(1-2), 36-44.

***Figures** are fine as are **Supplemental files**.*

Thank you. We have kept the Figures and Supplemental files unaltered unless changes were required to address the reviewer comments as detailed above.

Supplementary Table 4 A-F: Taxonomic alignment of representative taxa of key functional groups detected in BBD lesion communities in the natural environment with taxa found in the present study. Taxa found in natural BBD communities are shaded in grey. Corresponding taxa detected in the present study down to the resolved taxonomic level are shaded in green. *Cyanobacteriota* containing taxa with nitrogen fixation potential are highlighted in orange. The “Comment” column provides further information. The Ctr, Ht and BBDI columns indicate in which sample types the taxa were identified in the present study. Ctr = control colonies, Ht = Visually healthy tissue on BBD diseased colonies, BBD = Microbial mats from BBD lesions

A Cyanobacteriota (mat / habitat forming, creation of microenvironment)								
Phylum	Class	Order	Family	Genus	Ctr	Ht	BBD	Comment
Cyanobacteriota	Cyanobacteriia / Cyanophyceae	Cyanobacteriales	Desertifilaceae	Oscillatoria Some species fix nitrogen ^{1, 2, 3, 4, 5}				BBD in Bahamas, St Croix, Philippines, and Florida Keys ^{6, 7} Pre-BBD/BBD lesions Great Barrier Reef ⁸
Cyanobacteriota	Cyanobacteriia / Cyanophyceae	Cyanobacteriales	Desertifilaceae	Oscillatoria		x		
				Incertae Sedis_507	x	x	x	
Cyanobacteriota	Cyanobacteriia / Cyanophyceae	Cyanobacteriales	Desertifilaceae Some taxa fix nitrogen ⁴	Hormoscilla				Belize BBD ^{7, 9} Related to Roseofilum
Cyanobacteriota	Cyanobacteriia / Cyanophyceae	Cyanobacteriales	Desertifilaceae	Hormoscilla	x	x	x	BBD lesions, ASV325 top BLAST match (99.75%) with Uncultured bacterium clone from BBD lesions in Favia sp. corals from Eilat.
Cyanobacteriota	Cyanobacteriia / Cyanophyceae	Cyanobacteriales	Desertifilaceae Some taxa fix nitrogen ⁴	Roseofilum Some species fix nitrogen ¹⁰				BBD in Sidastrea from Belize/Florida ⁷ Roseofilum/Phormidium corallyticum is not always present in BBD mats. Instead, the dominant cyanobacteria in the mat are context dependent ^{11, 12} .
Cyanobacteriota	Cyanobacteriia / Cyanophyceae	Cyanobacteriales	Desertifilaceae	Not detected	x	x	x	
Cyanobacteriota	Cyanobacteriia / Cyanophyceae	Cyanobacteriales	Geitlerinemaceae	Geitlerinema Some species fix nitrogen ¹³				BBD in Bahamas and Florida Keys ^{6, 7}
Cyanobacteriota	Cyanobacteriia / Cyanophyceae	Cyanobacteriales	Geitlerinemaceae	Geitlerinema	x			

Cyanobacteriota	Sericytochromatia	Leptolyngbyales	Leptolyngbyaceae	Leptolyngbya				BBD in Bahamas, Philippines, St. Croix and Florida Keys ^{6,7}
			Some taxa fix nitrogen ⁴	Some species fix nitrogen ⁴				
Cyanobacteriota	Sericytochromatia	Leptolyngbyales	Leptolyngbyaceae	Leptolyngbya		x		
Cyanobacteriota	Sericytochromatia	Leptolyngbyales	Leptolyngbyaceae	Incertae Sedis_514	x	x		
Cyanobacteriota	Sericytochromatia_2920	Phormidesmiales	Phormidesmiaceae	Phormidium				BBD in Bahamas and Florida Keys ⁷
				Some species fix nitrogen ³				
Cyanobacteriota	Sericytochromatia_2920	Phormidesmiales	Phormidesmiaceae	Phormidium	x	x	x	BBD diseased colony, ASV6242 has a top BLAST match (100%) with Filamentous cyanobacterium FLK9 from BBD lesions taken from corals in the Northern Florida Keys ¹⁴ . BBD lesion, ASV14894 has a 98.01% sequence match with Cyanobacterium sp. BBD-AO-Green from black band disease from corals (Buerger et al., unpublished, see GenBank KU720413.1).
Cyanobacteriota	Cyanobacteriia / Cyanophyceae	Cyanobacteriales	Paraspirulinaceae	Spirulina				BBD, Microscopic ID ^{14, 17, 18}
		Some taxa fix nitrogen ^{15, 16}						
Cyanobacteriota	Cyanobacteriia / Cyanophyceae	Cyanobacteriales	Paraspirulinaceae	Spirulina	x	x	x	BBD lesions, ASV43 has a 99.01% sequence match with Cyanobacterium BBT from BBD bacterial mats (Frias-Lopez et al., unpublished, see GenBank AY515014.1). BBD lesions, ASV182 has a 99.26% ssequence match with Uncultured bacterium clone CD02013E12 from a BD mat on a Gorgonia ventalina colony from Curacao ¹⁹ .
			Xenococcaceae	unclassified_Xenococcaceae	x	x	x	
			Xenococcaceae	Pleurocapsa PCC-7319	x	x	x	
			Xenococcaceae	Chroococciopsis PCC-6712	x	x	x	
			Nostocaceae	Rivularia PCC-7116	x			
				Some species fix nitrogen ¹⁶				

			Nostocaceae	Mastigocoleus BC008	x	x		
				Some species fix nitrogen ¹⁵				
			Cyanobacteriaceae	Geminobacterium	x	x	x	
			Cyanobacteriaceae	Symphothece PCC-7002	x			
			Cyanobacteriaceae	Merismopedia AICB1015		x		
			Cyanobacteriaceae	Annamia HOs24	x	x		
			Coleofasciculaceae	Caldora VP642b	x			
Cyanobacteriota	Cyanobacteria / Cyanophyceae	Eurycoccales	Eurycoccales Incertae Sedis	Synechococcus				BBD in Great Barrier Reef Montipora ⁸
				Some species fix nitrogen ^{5, 20, 21}				
Cyanobacteriota	Cyanobacteria / Cyanophyceae	Eurycoccales	Eurycoccales Incertae Sedis	Synechococcus	x	x	x	
Cyanobacteriota	Sericytochromatia_424	Limnotrichales_2	Limnotrichaceae_2	Limnothrix				Pre-BBD lesions Great Barrier Reef ⁸
Cyanobacteriota	Sericytochromatia_424	Limnotrichales_2	Limnotrichaceae_2	Limnothrix	x	x		
Cyanobacteriota	Sericytochromatia_424	Pseudanabaenales_2	Pseudanabaenaceae_2	Pseudanabaena				BBD Red Sea ²²
				Some species fix nitrogen ³				
Cyanobacteriota	Sericytochromatia_424	Pseudanabaenales_2	Pseudanabaenaceae_2	Pseudanabaena	x	x		
		Obscuribacterales	Obscuribacteraceae	Incertae Sedis_528	x	x	x	
		Phormidesmiales	Nodosilineaceae	MBIC10086	x	x	x	BBD lesions, ASV153 found in BBD had a top BLAST match (100%) with Uncultured bacterium clone SGUS386 from White Plague Disease associated communities in Montastraea faveolata from the Caribbean ²³ .
		Phormidesmiales	Nodosilineaceae	unclassified_Nodosilineaceae		x	x	BBD diseased colony, ASV10931 had a 100% BLAST sequence match with Uncultured cyanobacterium clone Feb_09_5S.7 from BBD consortia from Favia sp. in the Red Sea ²⁴ .
		Phormidesmiales	Phormidesmiaceae	Phormidesmis	x	x	x	
		Phormidesmiales	Phormidesmiaceae	Acrophormium	x	x	x	
		Thermosynechococcales	Acaryochloridaceae	Acaryochloris	x	x	x	

Cyanobacteriota	Sericytochromatia_6194	Synechococcales Some taxa fix nitrogen ³	Synechococcales Incertae Sedis	Schizothrix Some species fix nitrogen ²⁵				Microscopic identification ^{14, 26}
Cyanobacteriota	Sericytochromatia_6194	Synechococcales	Synechococcales Incertae Sedis	Schizothrix	x	x	x	BBD diseased colony, ASV4183 99.75% BLAST match with Uncultured cyanobacterium clone MD3.8 from diseased colonies of Montastrea faveolata from the Caribbean (Johnson et al. unpublished, see GenBank FJ425596.1).
			Cyanobiaceae	Cyanobium PCC-6307	x	x	x	BBD lesions, ASV2092 99.5% sequence match with Uncultured cyanobacterium clone FRSSCT_14f from BBD coral tissue of Siderastrea siderea from St. Croix ²⁷ .
	Vampirivibronia	Caenarcaniphilales	Incertae Sedis_235	Incertae Sedis_526	x	x	x	
	Vampirivibronia	Gastranaerophilales	Gastranaerophilaceae	Incertae Sedis_527	x			

B Sulfate-reducing bacteria (Production of toxic sulfide)								
Phylum	Class	Order	Family	Genus	Ctr	Ht	BBD	Comment
Thermo desulfobacteriota	Desulfobacteria	Desulfobacterales	Desulfobacteraceae	Desulfobacter				BBD in corals in Curacao¹⁹ Increased in bleached corals from South China Sea²⁸
Thermo desulfobacteriota	Desulfobacteria	Desulfobacterales	Desulfobacteraceae	Desulfobacter	x	x	x	
				unclassified_Desulfobacteraceae	x	x	x	
			Desulfosarcinaceae	Desulfosarcina			x	
			Incertae Sedis_481	Incertae Sedis_999		x		
			Desulfolunaceae	unclassified_Desulfolunaceae		x		
			Desulfosarcinaceae	Incertae Sedis_998			x	BBD lesions , ASV18520 has a 100% BLAST sequence match with Uncultured delta proteobacterium clone D10 from anaerobic marine sediments (Koachling et al., unpublished, see GenBank GQ249552).
Thermodesulfobacteriota	Desulfovibrionia	Desulfovibrionales	Desulfovibrionaceae	Desulfovibrio				BBD in Red Sea^{22, 29} BBD on multiple Caribbean coral species³⁰
Thermodesulfobacteriota	Desulfovibrionia	Desulfovibrionales	Desulfovibrionaceae	Pseudodesulfovibrio	x	x	x	
				Halodesulfovibrio	x	x	x	BBD lesions , ASV5642 has a 99.77% sequence match with Uncultured bacterium clone BBD4-907-79 from BBD mat from a Favia sp. coral from the Red Sea, Israel ³¹ . BBD lesions , ASV5642 has a 99.77% sequence match with Uncultured bacterium clone BBD-Aug08-4BB-79 from BBD mat from a Favia sp. coral from the Red Sea, Israel (Arotsker et al., unpublished, see GenBank GU472053.1). BBD lesions , ASV5642 also has a 99.77% sequence match with Delta proteobacterium BBD 3-1 from BBD from Diplora clivosa from Florida (Viehman et al., unpublished, see GenBank AY750147.1).

										BBD lesions, ASV1431 has a 100% BLAST sequence match with Uncultured Desulfovibrio sp. clone MD3.10 found in diseased colonies of the Caribbean coral Montastrea faveolata (Johnson et al., unpublished, see GenBank FJ425598.1). BBD infected coral, ASV4061 has a 100% sequence match with Uncultured bacterium clone BBD-Aug08-4BB-106 found in a BBD affected Favia sp. coral from Eilat (Arotsker et al., unpublished, see GenBank GU472076.1). BBD infected coral, ASV4061 also has a 100% sequence match with Delta proteobacterium BBD 11-2 from BBD lesions from a Siderastrea siderea coral from Dominica (Viehman et al., unpublished, see GenBank AY750148). BBD lesions, ASV9131 has a 99.77% sequence match with Uncultured Desulfovibrio sp. clone MD3.10 from diseased colonies of Montastraea faveolata from the Caribbean (Johnson et al. unpublished, see GenBank FJ425598.1). BBD lesions, ASV9131 has a 99.77% sequence match with Uncultured bacterium clone SGUS1499 from diseased corals of Montastraea faveolata²³. BBD infected colony, ASV4061 has a 100% sequence match with Uncultured bacterium clone BBD-Aug08-4BB-106 from BBD affected Favia sp. corals from Eilat, Israel (Arotsker et al., unpublished, see GenBank GU472076.1). BBD infected colony, ASV4061 has a 100% sequence match with Delta proteobacterium BBD 11-2 from BBD of Siderastrea siderea from Dominica (Viehman et al., unpublished, see GenBank AY750148.1). BBD infected colony, ASV5177 has a 99.53% sequence match with Uncultured bacterium clone BBD4-907-79 from BBD mats from Favia sp. from the Red Sea, Israel³¹.
--	--	--	--	--	--	--	--	--	--	---

				Maridesulfovibrio	x			
				unclassified_Desulfovibrionaceae	x			
				Incertae Sedis_1006	x			
				Incertae Sedis_876	x	x	x	
Thermodesulfobacteriota	Desulfobulbia	Desulfobulbales	Desulfobulbaceae	Desulfobulbus				Increased in bleached corals from South China Sea²⁸
Thermodesulfobacteriota	Desulfobulbia	Desulfobulbales	Desulfobulbaceae	unclassified_Desulfobulbaceae	x		x	

C Sulfide-oxidizing bacteria (Maintaining the sulphur metabolism)									
Phylum	Class	Order	Family	Genus	Ctr	Ht	BBD	Comment	
Pseudomonadota	Gammaproteobacteria	Beggiatoales	Beggiatoaceae	Beggiatoa				BBD in the Red Sea ²² BBD in the wider Caribbean ^{7, 32} BBD in Bermuda ¹⁸	
Pseudomonadota	Gammaproteobacteria	Beggiatoales	Beggiatoaceae	Unclassified Beggiatoaceae	x	x	x	BBD lesions, Beggiatoaceae are Nitrate-dependent sulfide oxidizing bacteria ³³	
Pseudomonadota	Gammaproteobacteria	Beggiatoales	Beggiatoaceae	Thioflexithrix	x	x	x	BBD lesions, Sulfide-oxidising genus. ASV478 has a 100% sequence match with Beggiatoa sp. 35Flor from BBD lesions in a Scleractinian coral from the Florida Keys ³⁴ . Filamentous, sulfide-oxidizing bacterium ³⁵ .	
Campylobacterota	Campylobacteria	Campylobacterales	Arcobacteraceae	Arcobacter				BBD in the Red Sea ²² BBD in corals in Curacao ¹⁹	
Campylobacterota	Campylobacteria	Campylobacterales	Arcobacteraceae	Arcobacter	x	x	x	BBD lesions, ASV31 has a 100% sequence match with Uncultured bacterium clone CD02013E10 found in BBD mats of Gorgonia ventalina ¹⁹ BBD lesions, ASV18 has a 100% sequence match with Uncultured bacterium clone SHFG542 from diseased tissue (White Plague Disease) of Montastraea faveolate from the Caribbean ²³ .	
Campylobacterota	Campylobacteria	Campylobacterales	Sulfurospirillaceae	Sulfurospirillum				BBD in the Red Sea ²²	
Campylobacterota	Campylobacteria	Campylobacterales	Sulfurospirillaceae	Sulfurospirillum	x				
Campylobacterota	Campylobacteria	Campylobacterales	Sulfurimonadaceae	Sulfurimonas	x	x		Sulfide-oxidising genus Sulfurimonas can reduce nitrate to nitrite or N ₂ , enabling anaerobic growth via denitrification ³⁶ .	

D Heterotrophic & opportunistic bacteria									
Phylum	Class	Order	Family	Genus	Ctr	Ht	BBD	Comment	
Pseudomonadota	Gammaproteobacteria	Enterobacterales	Vibrionaceae	Vibrio				BBD-infected corals in the Red Sea²⁹ Vibrio isolated from BBD bands, culture and proteolytic activity tested³⁷ Vibrio isolated from BBD bands in the wider Caribbean³⁸	
Pseudomonadota	Gammaproteobacteria	Enterobacterales	Vibrionaceae	Vibrio	x	x	x	BBD lesions, ASV1406 has a 100% sequence match with Vibrio sp. BD6B from the sulphurus layer of a diseased Montipora aequituberculata skeleton with White Syndrome³⁹.	
				Unclassified Vibrionaceae	x	x	x	BBD lesions, ASV45 has a 100% sequence match with Vibrio Pelagius strain 5100744_CD3_3 from diseased corals with White Syndrome from Malaysia (Akmal et al., unpublished, see GenBank PP980502.1) and a 100% sequence match with Vibrio sp. 4H1 from a Montastraea faveolata coral with Yellow Band Disease from Puerto Rico (Cunning et al., unpublished, see GenBank EU517650.1).	
				Photobacterium	x	x	x		
				Grimontia	x				
				Salinivibrio	x		x		
				Aliivibrio	x				
				Enterovibrio	x	x			
				Catenococcus_2	x		x		
Pseudomonadota	Gammaproteobacteria	Enterobacterales	Pseudoalteromonadaceae	Pseudoalteromonas				Isolates from BBD bands in the wider Caribbean⁴⁰ Identified in BBD bands from Indonesia^{41, 42}	

Pseudomonadota	Gammaproteobacteria	Enterobacterales	Pseudoalteromonadaceae	Pseudoalteromonas	x	x	x	BBD lesions , ASV2320 has a 100% sequence match with Pseudoalteromonas sp. Pad1.13 from diseased gorgonian Pseudopterogorgia americana (Vizcaino et al., unpublished, see GenBank GQ406581.1).
				Algicola	x	x	x	BBD lesions , ASV48 has a 100% sequence match on BLAST with Uncultured bacterium clone R3AE3C39 found in the Porites White Patch Syndrome-infected tissue of corals from South Africa (Sere et al., unpublished, see GenBank KF180023.1). BBD lesions , ASV1177 has a 99.77% sequence match with Uncultured bacterium clone Thai19_H05 from White Plague Disease infected coral Pavona duerdeni ⁴³ .
				unclassified_Pseudoalteromonadaceae	x	x	x	
				Psychrosphaera	x		x	
				Incertae Sedis_894	x	x		
Pseudomonadota	Gammaproteobacteria	Enterobacterales	Alteromonadaceae	Alteromonas				BBD mats in the Red Sea ²² BBD mats in the wider Caribbean ⁴⁴ BBD mats in the GBR ⁸
Pseudomonadota	Gammaproteobacteria	Enterobacterales	Alteromonadaceae	Alteromonas	x	x	x	
				Incertae Sedis_880	x	x	x	
				unclassified_Alteromonadaceae_2	x	x	x	
				Aestuariusbacter		x	x	
				Marisediminitalia	x			
				Planctobacterium	x	x	x	
				Glaciecola	x	x		
				Neptunicella	x			
				Lacimicrobium	x			

				Aliiglaciecola	x	x		
				Paraglaciecola	x			
Pseudomonadota	Alphaproteobacteria	Rhodobacterales	Paracoccaceae	Ruegeria				BBD mats in the wider Caribbean^{7, 11, 32} BBD mats in Okinawa, Japan⁴⁵
Pseudomonadota	Alphaproteobacteria	Rhodobacterales	Paracoccaceae	Ruegeria	x	x	x	BBD lesions , ASV17 has a 100% sequence BLAST match with Uncultured bacterium clone BBD-Aug08-3BB-31 from BBD affected Favia sp. corals in Eilat, Israel (Arotsker et al., unpublished, see GenBank GU472124.1). BBD lesions , ASV138 has a 99.75% sequence match with Uncultured bacterium clone BBDS16S-15 from BBD mats from Favites sp. from the Red Sea ²⁹ . BBD lesions , ASV752 has a 100% sequence match with Uncultured bacterium clone SHFG523 from diseased tissue of White Plague Disease infected Montastraea faveolata from the Caribbean ²³ .
				unclassified_Paracoccaceae	x	x	x	BBD lesions , ASV9 has a 100% sequence match with Uncultured alpha proteobacterium clone BBD_217_23 from BBD diseased coral tissue of Siderastrea siderea from Florida ⁴⁶ . BBD lesions , ASV15 has a 100% sequence match with Uncultured bacterium clone SHFG549 from diseased tissue of Montastraea faveolata infected with White Plague Disease from the Caribbean ²³ .
				Shimia	x	x	x	BBD lesions , ASV64 has a 100% sequence match with Uncultured bacterium clone BBD-Aug08-1BB-18 from BBD affected coral Favia sp. in Eilat, Israel (Arotsker et al., unpublished, see GenBank GU472113.1).
				Cognatishimia	x	x	x	BBD lesions , ASV76 has a 99.25% sequence match with Uncultured alpha proteobacterium clone BBD_HS216b_07 found in BBD coral tissues of Siderastrea siderea in the Bahamas (Richardson et al., unpublished, see GenBank DQ644016.1).
				Dinoroseobacter	x	x	x	

				Roseobacter clade CHAB-I-5 lineage	x	x	x	BBD lesions , ASV300 has a 100% sequence match with Uncultured bacterium clone SGUS600 found in Montasraea faveolata study looking at White Plaque Disease ²³ .
				Pikeienuella	x	x	x	
				Marivita	x	x	x	
				Silicimonas	x	x	x	BBD lesions , ASV390 has a 99.75% sequence match with Uncultured alpha proteobacterium B7-20G from Porites lobata with BBD from the Great Barrier Reef (Cooney et al., unpublished, see GenBank AY148320.1).
				Actibacterium	x	x	x	BBD lesions , ASV808 has a 99.75% sequence match with Uncultured bacterium clone SGUS1403 found in Montasraea faveolata study looking at White Plaque Disease ²³ .
				Limibaculum	x	x	x	
				Tropicibacter	x		x	BBD lesions , ASV1316 has a 99.5% sequence match with Uncultured bacterium clone BBDS16S from BBD mats from Favites sp. from the Red Sea ²⁹ .
				Roseovarius	x	x	x	
				Sulfitobacter	x			Found in early stages of BBD infection in corals ⁸ .
				Rubrimonas	x	x		
				Tropicibacter	x		x	BBD lesions , ASV1316 has a 99.5% sequence match with Uncultured bacterium clone BBDS16S-10 from BBD mats from Favites sp. in the Red Sea ²⁹ .
				Yoonia	x	x	x	BBD lesions , ASV2636 has a 99.25% sequence match with Uncultured alpha proteobacterium clone CL18-G05 from a cyanobacterial patch stage of BBD development in Montipora hispida from the Great Barrier Reef ⁸ .
				Tateyamarina	x		x	
				Tranquillimonas	x	x	x	
				Muriiphilus		x		

				Incertae Sedis_811	x	x	x	BBD lesions , ASV10716 is a 100% sequence match with Uncultured bacterium clone BB31NT16S-9 found in Favites sp. mucus adjacent to BBD mat in the Red Sea ²⁹ .
				Allosediminivita	x	x		
				Lentibacter	x	x		
				Limimanicola	x			
				Planktomarina	x			
				Maribius	x			
				Loktanella	x			
				Chachezhania		x		
Pseudomonadota	Gammaproteobacteria	Pseudomonadales	Marinobacteraceae	Marinobacter				BBD mats in the Bahamas⁴⁷ BBD mats in the wider Caribbean⁴⁸
Pseudomonadota	Gammaproteobacteria	Pseudomonadales	Marinobacteraceae	Marinobacter	x	x		
Bacteroidota	Bacteroidia	Flavobacteriales	Flavobacteriaceae	Flavobacterium				BBD mats in the wider Caribbean^{11, 32}
Bacteroidota	Bacteroidia	Flavobacteriales	Flavobacteriaceae	Flavobacterium	x	x		
				Winogradskyella	x	x	x	BBD lesions , ASV335 has a 100% sequence match with Uncultured Winogradskyella sp. clone MD3.56 in diseased Caribbean coral Montastrea faveolate (Johnson et al., unpublished, see GenBankFJ425644.1).
				Tenacibaculum	x	x	x	
				Pseudotenacibaculum	x	x	x	

E Recurrent secondary genera detected in BBD sequencing reports								
Phylum	Class	Order	Family	Genus	Ctr	Ht	BBD	Comment
Spirochaetota	Sericytochromatia_6043	Spirochaetales	Spirochaetaceae	Spirochaeta				Red Sea BBD ²² Caribbean BBD ³²
Spirochaetota	Sericytochromatia_6043	Spirochaetales	Spirochaetaceae	Spirochaeta	x	x	x	BBD lesions , ASV8588 has a 100% sequence match with Uncultured Spirochaetes bacterium clone 1HP1-B8 from a Turbinaria mesenterina colony affected by White Syndrome in Australia (Godwin et al., unpublished, see GenBank EU780318.1).
				Sediminspirochaeta	x	x	x	
				Oceanispirochaeta	x		x	
				Thiospirochaeta	x			
Bacillota	Sericytochromatia_2834	Peptostreptococcales-Tissierellales	Fusibacteraceae	Fusibacter				Red Sea BBD ²² Caribbean BBD ^{32, 44, 48} BBD wider Caribbean ⁴⁹
Bacillota	Sericytochromatia_2834	Peptostreptococcales-Tissierellales	Fusibacteraceae	Fusibacter	x	x	x	BBD lesions , ASV25 has a 100% sequence match with Uncultured bacterium clone BBD4-907-2 from a BBD mat from Favia sp. in Eilat, Israel ³¹ . BBD lesions , ASV63 has a 99.75% sequence match with Uncultured bacterium clone BBD4-907-2 from BBD mats on infected Favia sp. corals from the Red Sea ³¹ . BBD lesions , ASV63 has a 99.75% sequence match with Uncultured Firmicutes bacterium clone WA_20bf from BBD infected coral tissues from Siderastrea sidereal corals from the wider Caribbean ³² . BBD lesions , ASV99 has a 100% sequence match with Uncultured bacterium clone CD02002D01 from BBD mat from Colpophyllia natans from Curacao ¹⁹ . BBD lesions , ASV374 has a 99.75% sequence match with Uncultured bacterium

								clone BB3S16S-1 from BBD mat from Favites sp. corals from the Red Sea ²⁹ .
Bacillota	Clostridia	Clostridiales	Clostridiaceae	Clostridium				Red Sea BBD²² GBR BBD⁸ Caribbean BBD⁴⁴
Bacillota	Clostridia	Clostridiales	Clostridiaceae	Clostridium	x			
				unclassified_Clostridiaceae	x			
				Oceanirhabdus	x			
				Incertae Sedis_220	x			

Supplementary Table 5: Compilation of temperature and nutrient data retrieved from NOAA coral watch and Bio-Oracle data bases. Locations are ordered alphabetically by the first author's name of the corresponding reference (Table S1). NA= No data available

Location Name (reference)	Location Latitude	Location Longitude	Temperature monitoring Start	Temperature monitoring End	Maximum DHW (NOAA)	DHW Category [weeks]	NO ₃ [mol*m ⁻³] (Bio-Oracle)	PO ₄ [mol*m ⁻³] (Bio-Oracle)	N:P (Bio-Oracle)
aeby2015	22.228688	-159.45599	31/10/2011	31/10/2012	0	<4	NA	NA	NA
aeby2020	25.60133	56.352383	31/08/2015	31/08/2016	0.52	<4	0.064319345	0.153860279	0.41803735
aeby2020PG	24.599	54.4215	31/08/2015	31/08/2016	0.15	<4	0.00905256	0.007359598	1.23003447
aeby2020SOH	26.241445	56.197121	31/08/2015	31/08/2016	0.31	<4	0.016749272	0.08614569	0.19442959
aeby2021	20.162507	40.101479	30/11/2014	30/11/2015	8.35	>8	0.006575177	0.009401585	0.69936895
ainsworth2007	29.501366	34.917253	30/06/2004	30/06/2005	NA	Undetermined	NA	NA	NA
arotsker2015	29.49853	34.92861	31/08/2008	31/08/2009	1.02	<4	0.024551077	0.005397038	4.5489912
arotsker2016	29.49853	34.92861	31/08/2011	31/08/2012	4.11	4 to 8	0.024551077	0.005397038	4.5489912
arotssker2009	29.49473	34.92692	31/10/2006	31/10/2007	2.2	<4	0.024551077	0.005397038	4.5489912
barneah200705	29.496829	34.92047	31/11/2004	31/11/2005	NA	Undetermined	0.024551077	0.005397038	4.5489912
barneah200706	29.496829	34.92047	31/05/2005	31/05/2006	0.8	<4	0.024551077	0.005397038	4.5489912
bastidas2012	11.866014	-66.781413	NA	NA	NA	Undetermined	0.078995817	0.006296	12.5469847
bhedi2017	25.14665	-80.292983	NA	NA	NA	Undetermined	0.111252753	0.000759836	146.416794
borger2003C	15.443636	-61.459121	31/08/1999	31/08/2000	1.96	<4	0.013105321	0.001291926	10.1440191
borger2003M	15.381714	-61.425086	31/08/1999	31/08/2000	1.75	<4	0.015151726	0.001286541	11.7771063
borger2003R	15.306115	-61.400138	31/08/1999	31/08/2000	1.58	<4	0.016033694	0.001285603	12.4717321
borger2003S	15.42545	-61.444439	31/08/1999	31/08/2000	NA	Undetermined	NA	NA	NA
borger2003SO	15.230977	-61.376417	31/08/1999	31/08/2000	1.75	<4	0.016242291	0.001286919	12.6210651
borger200501	25.382933	-80.157749	30/05/2000	30/05/2001	0	<4	0.10228806	0.001072937	95.3346024
borger200502	25.382933	-80.157749	30/05/2001	30/05/2002	0	<4	0.10228806	0.001072937	95.3346024
bourne2011	-18.55	146.5	NA	NA	NA	Undetermined	0.044077367	0.08819787	0.49975547
bozett200703	-14.66667	145.4667	31/07/2002	31/07/2003	0.93	<4	0.003169784	0.139892624	0.02265869

boyett200704	-14.66667	145.4667	31/05/2003	31/05/2004	2.5	<4	0.003169784	0.139892624	0.02265869
brandt2009	24.89553	-80.62773	31/10/2004	31/10/2005	5.49	4 to 8	0.154888037	0.000607614	254.911865
buerger2016	-18.57682	146.4966	30/06/2012	30/06/2013	2.03	<4	0.044077367	0.08819787	0.49975547
buerger2019	-18.6376	146.4982	31/07/2007	31/07/2008	0	<4	0.044459577	0.088435257	0.50273591
casey2014	-14.68472	145.4486	31/08/2012	31/08/2013	1.47	<4	0.004354826	0.138492355	0.03144453
chong2011CH	-14.686611	145.460041	28/02/2008	28/02/2009	1.69	<4	0.003169784	0.139892624	0.02265869
chong2011HS	-14.687515	145.444014	28/02/2008	28/02/2009	1.67	<4	0.004354826	0.138492355	0.03144453
chong2011LB	-14.685974	145.453798	28/02/2008	28/02/2009	1.69	<4	0.003169784	0.139892624	0.02265869
chong2011LH	-14.686796	145.471812	28/02/2008	28/02/2009	1.69	<4	0.003169784	0.139892624	0.02265869
chong2011NR	-14.645746	145.454991	28/02/2008	28/02/2009	1.56	<4	0.00310751	0.141506928	0.02196013
chong2011P	-14.690623	145.454256	28/02/2008	28/02/2009	1.69	<4	0.003169784	0.139892624	0.02265869
cooney2002	13.240759	-59.519151	30/05/1999	30/05/2000	1.52	<4	0.061673084	0.001494758	41.2595825
croquer2003	11.932889	-66.662055	31/08/1999	31/08/2000	6.67	4 to 8	0.082420747	0.006138537	13.4267731
das2020	26.644818	127.855169	NA	NA	NA	Undetermined	0.122153308	0.030482775	4.00728989
dorrestein2020	17.420523	-62.811944	31/05/2016	31/05/2017	1.97	<4	0.00232183	0.001525126	1.52238548
eaton2022	17.789987	-64.613198	31/07/2019	31/07/2020	5.4	4 to 8	0.004038347	0.001353989	2.98255491
frias2003CA00	12.103297	-68.957471	31/08/1999	31/08/2000	1.88	<4	0.020245312	0.010741401	1.88479257
frias2003CA01	12.103297	-68.957471	31/08/2000	31/08/2001	0	<4	0.020245312	0.010741401	1.88479257
frias2003CJ01	12.103297	-68.957471	31/01/2000	31/01/2001	0.16	<4	0.020245312	0.010741401	1.88479257
frias2003NB	-4.965984	151.127918	31/08/1999	31/08/2000	0.5	<4	0.076522887	0.175507222	0.43600991
garcia2016	-17.998094	-38.671336	28/02/2011	28/02/2012	0	<4	0.008607847	0.033906766	0.25386813
glas2012	-18.56	146.5017	31/03/2009	31/03/2010	1.24	<4	0.042037281	0.089408321	0.4701719
haapkyla2010	-23.43	151.999	31/08/2008	31/08/2009	0	<4	0.007817617	0.126405749	0.06184542
hadaidi2018AM	22.0766	38.7751	30/11/2014	30/11/2015	2.82	<4	0.018671589	0.017837018	1.04678869
hadaidi2018FR1	20.1732	40.1613	30/11/2014	30/11/2015	8.59	>8	0.006323713	0.009022856	0.7008549
hadaidi2018IF	22.2358	39.0304	30/11/2014	30/11/2015	4.13	4 to 8	0.015825022	0.013358437	1.18464613
hadaidi2018LP	21.7092	39.0832	30/11/2014	30/11/2015	NA	Undetermined	0.06053853	0.060449835	1.00146723

hadaidi2018QAK	20.1407	40.0931	30/11/2014	30/11/2015	8.05	>8	0.006983132	0.00978757	0.71346933
hadaidi2018SH	22.2012	38.9992	30/11/2014	30/11/2015	3.77	<4	0.016240858	0.013369129	1.21480298
hadaidi2018SR	19.8985	40.1514	30/11/2014	30/11/2015	6.89	4 to 8	0.005875924	0.009426739	0.62332517
hadaidi2018WR	20.123	40.2118	30/11/2014	30/11/2015	9.22	>8	0.00594359	0.008270533	0.71864659
haya2023Pad	-4.102197	121.468687	30/06/2021	30/06/2022	1	<4	0.054285924	0.014668821	3.70076942
haya2023Pas	-4.022342	122.751646	30/06/2021	30/06/2022	0	<4	0.104224221	0.035178991	2.96268368
haya2023W	-5.295956	123.49631	30/06/2021	30/06/2022	1.24	<4	0.003350721	0.043897757	0.07633012
hazraty2021k14	26.48274	53.98141	31/08/2013	31/08/2014	0	<4	0.012194254	0.051145275	0.23842385
hazraty2021k15	26.87529	56.32875	31/08/2014	31/08/2015	1.43	<4	0.007410078	0.093931073	0.07888847
henao2017	11.25693	-74.23143	NA	NA	NA	Undetermined	0.621453916	0.007451991	83.3943481
huang2021GI	22.642501	121.482283	30/06/2017	30/06/2018	3.83	<4	0.081652557	0.052830857	1.54554677
huang2021K	21.92294	120.75259	30/06/2017	30/06/2018	1.84	<4	0.091929383	0.048611596	1.89109981
huang2021SI	23.218879	119.410285	NA	NA	NA	Undetermined	0.446895659	0.039682091	11.261898
johan2014BL	-2.833199	108.411491	31/03/2013	31/03/2014	0.45	<4	0.017924487	0.000978	18.327692
johan2014M	-2.637231	108.487644	31/07/2013	2014-04-31	0.45	<4	0.019515161	0.001209665	16.1326942
johan2016EP	-5.858472	106.623861	NA	NA	NA	Undetermined	0.29660103	0.279253945	1.06211936
johan2016NP	-5.740249	106.620694	NA	NA	NA	Undetermined	0.324007791	0.108960441	2.97362781
johan2016P	-5.452111	106.562472	NA	NA	NA	Undetermined	0.349904491	0.008230903	42.5110664
johan2016PT	-5.459444	106.563694	NA	NA	NA	Undetermined	0.349904491	0.008230903	42.5110664
johan2016SP	-5.870556	106.610778	NA	NA	NA	Undetermined	0.29660103	0.279253945	1.06211936
johan2016Z2SP	-5.750528	106.611528	NA	NA	NA	Undetermined	0.320774282	0.131996872	2.43016577
jones201204	32.304528	-64.755069	31/07/2003	31/07/2004	6.96	4 to 8	0.354073925	0.03002596	11.7922602
jones201205	32.304528	-64.755069	31/07/2004	31/07/2005	0.44	<4	0.354073925	0.03002596	11.7922602
jones201206	32.304528	-64.755069	31/07/2005	31/07/2006	1.43	<4	0.354073925	0.03002596	11.7922602
jones201207	32.304528	-64.755069	31/07/2006	31/07/2007	0	<4	0.354073925	0.03002596	11.7922602
jones201208	32.304528	-64.755069	31/07/2007	31/07/2008	1.16	<4	0.354073925	0.03002596	11.7922602
jordan2005CA	20.208052	-91.969759	NA	NA	NA	Undetermined	0.054796065	0.000747718	73.2843933

jordan2005PM	20.845331	-86.858996	NA	NA	NA	Undetermined	0.025771562	0.005831977	4.41900969
jordan2005TE	20.914783	-92.212866	NA	NA	NA	Undetermined	0.047469079	0.00084562	56.1352615
kaczmarzsky2005BB	18.253	-65.47467	30/09/2000	30/09/2001	0.31	<4	0.002750306	0.000957001	2.87387991
kaczmarzsky2005F	18.23333	-65.455	30/09/2000	30/09/2001	0.3	<4	0.00273938	0.000963594	2.84287691
kaczmarzsky2006	10.211897	118.912402	31/08/2003	31/08/2003	0	<4	0.010808201	0.008697049	1.24274349
klaus2011	12.109439	-68.954585	30/06/2003	31/07/2004	8.11	>8	0.020245312	0.010741401	1.88479257
kuehl2011	32.458969	-64.831507	31/07/2007	30/06/2008	0.81	<4	0.368066281	0.030927157	11.9010706
lamb2011A	-11.47157	147.020202	31/07/2008	31/07/2009	5.83	4 to 8	0.001848947	0.132797016	0.01392311
lamb2011B	-13.91632	148.433625	31/07/2008	31/07/2009	7.59	4 to 8	0.001547558	0.168721084	0.00917228
lamb2011C	-17.13060	152.113122	31/07/2008	31/07/2009	9.22	>8	0.000997517	0.146927771	0.00678916
lamb2011D	-17.94671	151.17383	31/07/2008	31/07/2009	3.97	<4	0.001117534	0.133529326	0.0083692
lamb2014	10.065142	99.852992	NA	NA	NA	Undetermined	0.263282641	0.000281987	933.667847
lewis2017	25.37792	-80.178102	NA	NA	NA	Undetermined	0.10228806	0.001072937	95.3346024
lewis201732	24.476774	-81.745892	NA	NA	NA	Undetermined	0.224282007	0.000401885	558.075562
lewis2017C	24.683856	-80.967144	NA	NA	NA	Undetermined	0.186354668	0.000473201	393.817047
lewis2017CO	24.588242	-81.272968	NA	NA	NA	Undetermined	0.16884407	0.000634468	266.118988
lewis2017P	25.008693	-80.383181	NA	NA	NA	Undetermined	0.132011904	0.000679685	194.225082
lewis2017S1	24.458025	-81.873608	NA	NA	NA	Undetermined	0.231348744	0.000451066	512.893555
lewis2017S2	24.450524	-81.940899	NA	NA	NA	Undetermined	0.238695229	0.000432513	551.880066
meyer2016B	16.8028	-88.0825	30/04/2013	30/04/2014	1.24	<4	0.032938226	0.000606476	54.3108139
meyer2016FL	24.5472	-81.3811	30/04/2013	30/04/2014	0	<4	0.172069521	0.00069348	248.124634
meyer2016H	15.837	-83.292	30/04/2013	30/04/2014	2.12	<4	0.022094316	0.000155358	142.215134
meyer2023	16.8028	-88.0825	30/09/2014	30/09/2015	2.59	<4	0.032938226	0.000606476	54.3108139
meyer2023fk	24.5472	-81.3811	31/07/2016	31/07/2017	2.19	<4	0.172069521	0.00069348	248.124634
meyer2023fl	26.09965	-80.09364	31/07/2017	31/07/2018	0.14	<4	0.094876949	0.003041845	31.1905918
mhuantong2016	-5.832694	110.3807	30/09/2014	30/09/2015	0	<4	0.023487901	0.002321483	10.11763
mhuantong2016K	-5.836917	110.4206	30/09/2014	30/09/2015	0	<4	0.0244202	0.002454673	9.94845486

miller2011	25.14667	-80.29333	NA	NA	NA	Undetermined	0.111252753	0.000759836	146.416794
montano2012	3.066667	72.95	30/11/2009	30/11/2010	0.77	<4	0.007518255	0.019178375	0.39201733
montano2013	4.066218	73.148198	31/10/2010	31/10/2011	0	<4	0.011351929	0.01814733	0.62554264
montano2013A	4.07	72.82	31/10/2010	31/10/2011	0	<4	0.009920737	0.01815184	0.54654163
montano2013F	3.2333	72.9333	31/10/2010	31/10/2011	0	<4	0.007528882	0.018874176	0.3988986
muller2011	17.76417	-64.61944	30/09/2008	30/09/2009	0.95	<4	0.004038347	0.001353989	2.98255491
muller2017	24.54794	-81.457	31/07/2012	31/07/2013	0.74	<4	0.190963037	0.000700729	272.520599
myers2007CR	24.961468	-80.453958	31/05/2004	31/05/2005	1.01	<4	0.152273766	0.000551392	276.162445
myers2007DL	24.927391	-80.501165	31/07/2005	31/07/2006	5.06	4 to 8	0.148883149	0.000594632	250.378494
myers2007FR	25.037455	-80.350132	31/10/2005	31/10/2006	2.97	<4	0.132011904	0.000679685	194.225082
myers2007MO	25.015008	-80.378541	30/06/2003	30/06/2004	0	<4	0.132011904	0.000679685	194.225082
myers2007RB	23.80169	-76.164775	31/07/2004	31/07/2005	0.32	<4	0.022166277	0.00172639	12.8396721
myers2007SP	23.774547	-76.095579	31/07/2004	31/07/2005	0.14	<4	0.01091445	0.002197115	4.96762705
myers2007USVI	18.23333	-65.455	31/08/2004	31/08/2005	2.17	<4	0.00273938	0.000963594	2.84287691
myers2007WB04	25.038181	-80.383034	30/06/2003	30/06/2004	0	<4	0.132011904	0.000679685	194.225082
myers2007WB06	25.038181	-80.383034	31/10/2005	31/10/2006	2.97	<4	0.132011904	0.000679685	194.225082
myers2009A	9.340773	123.31095	31/08/2004	31/08/2005	0	<4	0.146089133	0.026220184	5.57162905
myers2009F	18.23333	-65.455	30/06/2006	30/06/2007	4.32	4 to 8	0.00273938	0.000963594	2.84287691
myers2009HR	23.76967	-76.14117	31/08/2003	31/08/2004	0	<4	0.018845061	0.001855537	10.1561222
nicolet2018	-14.67431	145.435756	31/03/2022	31/03/2023	2.94	<4	0.004354826	0.138492355	0.03144453
oberle2019	22.232657	-159.56612	31/08/2015	31/08/2016	5.85	4 to 8	0.000730396	0.23295063	0.00313541
onton2011B	-21.87172	114.159002	31/01/2009	31/01/2010	2.24	<4	0.003756867	0.004394342	0.8549329
onton2011BB1	-23.14063	113.7418	31/05/2008	31/05/2009	3.47	<4	0.002353956	0.006306264	0.3732726
onton2011BB2	-23.14063	113.7418	31/01/2009	31/01/2010	2.45	<4	0.002353956	0.006306264	0.3732726
onton2011LB	-22.49902	113.705811	31/01/2009	31/01/2010	1.08	<4	0.014552543	0.007568326	1.92282176
onton2011NMI	-21.65605	114.369262	31/01/2009	31/01/2010	1.85	<4	0.003287608	0.004421744	0.74350953
onton2011SMI	-21.67623	114.344547	31/01/2009	31/01/2010	1.69	<4	0.00384097	0.00477919	0.80368626

onton2011W	-22.48966	113.708402	31/01/2009	31/01/2010	1.08	<4	0.014552543	0.007568326	1.92282176
page2006CB	-23.84444	151.310693	31/03/2003	31/03/2004	5.65	4 to 8	0.024485825	0.101269836	0.24178794
page2006LI	-14.80267	145.376607	31/03/2003	31/03/2004	2.56	<4	0.007040454	0.136029493	0.05175682
page2006TV	-19.04991	146.701038	31/03/2003	31/03/2004	1.53	<4	0.023370334	0.095836343	0.2438567
page2017	-20.82691	115.468979	NA	NA	NA	Undetermined	0.003234113	0.002493882	1.29681849
ponti2016AK	1.77691	125.20351	30/11/2012	30/11/2013	0	<4	0.002997905	0.023166315	0.12940793
ponti2016MF	1.762565	125.130998	30/11/2012	30/11/2013	0.15	<4	0.002884929	0.02272105	0.12697166
randal2014B	9.356201	-82.328193	30/09/2011	30/09/2012	1.99	<4	0.311309358	0.000586315	530.958801
randal2014M	18.710618	-87.675807	30/09/2011	30/09/2012	1.97	<4	0.011903049	0.001497663	7.94775105
randal2014SJ	18.356603	-64.774642	30/09/2011	30/09/2012	0.74	<4	0.002911812	0.001247373	2.33435655
randal2014T	20.962917	-97.297702	30/09/2011	30/09/2012	1.33	<4	0.654283598	0.001290036	507.182373
ranith2017A	10.836256	72.167881	31/12/2010	31/12/2011	0	<4	0.045717348	0.008803636	5.19300699
ranith2017B	10.93547	72.295005	31/12/2010	31/12/2011	0	<4	0.04905632	0.008675206	5.65477324
ranith2017C	11.691595	72.70569	31/12/2010	31/12/2011	0	<4	0.078685308	0.011227674	7.00815773
ranith2017K	10.090779	73.635065	31/12/2010	31/12/2011	0	<4	0.072139489	0.019322913	3.73336506
ranith2017KV	10.559656	72.624124	31/12/2010	31/12/2011	0	<4	0.05589238	0.00990153	5.64482212
ranith2017M	8.294013	73.040387	31/12/2010	31/12/2011	0	<4	0.044936891	0.010554282	4.25769281
ranith2017T	10.944788	72.321174	31/12/2010	31/12/2011	0	<4	0.049569706	0.008695843	5.70039082
rasoulouniriana2009	29.49427	34.92018	NA	NA	NA	Undetermined	0.024551077	0.005397038	4.5489912
richardson2007CO	24.961468	-80.453958	30/06/2003	30/06/2004	0.15	<4	0.152273766	0.000551392	276.162445
richardson2007DL	24.927391	-80.501165	30/06/2003	30/06/2004	0	<4	0.148883149	0.000594632	250.378494
richardson2007FR	18.23333	-65.455	31/07/2005	31/07/2006	9.92	>8	0.00273938	0.000963594	2.84287691
richardson200904	23.776022	-76.097083	31/08/2003	31/08/2004	0	<4	0.01091445	0.002197115	4.96762705
richardson200905	23.776022	-76.097083	30/07/2004	30/07/2005	0.14	<4	0.01091445	0.002197115	4.96762705
riegl2015	24.736133	54.584888	31/08/2010	31/08/2011	0	<4	0.008600598	0.005103954	1.68508542
rivera2018	20.904458	-86.839013	31/10/2016	31/10/2017	4.34	4 to 8	0.0227502	0.005992129	3.79668069
rodriguez2008	10.84167	-64.14724	31/08/2004	31/08/2005	14.04	>8	0.121058618	0.038079138	3.17913222

roff2008hr	-23.4513	151.957285	30/11/2003	30/11/2004	2.51	<4	0.006306826	0.126141767	0.04999792
roff2008wr	-23.46853	151.880999	30/11/2003	30/11/2004	2.99	<4	0.004806158	0.122685234	0.03917471
sato2009	-18.56	146.5017	31/01/2005	31/01/2006	1.06	<4	0.042037281	0.089408321	0.4701719
sato2010	-18.56	146.5017	31/08/2006	31/08/2007	0	<4	0.042037281	0.089408321	0.4701719
sato2010NE	-18.54167	146.5	31/08/2006	31/08/2007	0	<4	0.048054491	0.087067885	0.55191982
sato2011	-18.56	146.5017	31/01/2007	31/01/2008	0	<4	0.042037281	0.089408321	0.4701719
sato2017	-18.54167	146.5	30/11/2008	30/11/2009	0.29	<4	0.048054491	0.087067885	0.55191982
sekar2008BB	18.253	-65.47467	31/10/2004	31/10/2005	10.18	>8	0.002750306	0.000957001	2.87387991
sekar2008FK	25.15983	-80.25383	31/05/2004	31/05/2005	0.97	<4	0.110218507	0.000706541	155.997375
sekar2008FR	18.23333	-65.455	30/06/2004	30/06/2005	0.45	<4	0.00273938	0.000963594	2.84287691
sere2015M	-12.649	45.1033	31/03/2011	31/03/2012	0	<4	0.002860302	0.009623516	0.29722002
sere2015R	-21.0803	55.2195	31/12/2009	31/12/2010	0.48	<4			
sere2015SB	-27.3151	32.4111	28/02/2010	28/02/2011	NA	Undetermined			
sere2016	-21.087	55.19657	NA	NA	NA	Undetermined	0.001039467	0.00592643	0.17539509
sere20163C	-21.08676	55.19458	NA	NA	NA	Undetermined	0.001039467	0.00592643	0.17539509
sere2016LC	-21.16596	55.28508	NA	NA	NA	Undetermined	0.001219088	0.005804556	0.21002264
sere2016LC2	-21.16594	55.28193	NA	NA	NA	Undetermined	0.001219088	0.005804556	0.21002264
sere2016RDP	- 21.176397	55.285985	NA	NA	NA	Undetermined	0.001219088	0.005804556	0.21002264
sere2016RDP2	-21.17549	55.28346	NA	NA	NA	Undetermined	0.001219088	0.005804556	0.21002264
sere2016TD	- 21.103312	55.242294	NA	NA	NA	Undetermined	0.001112779	0.00586358	0.18977806
sere2016TD2	-21.10616	55.23954	NA	NA	NA	Undetermined	0.001112779	0.00586358	0.18977806
seveso2017	3.066667	72.95	31/10/2011	31/10/2012	0	<4	0.007518255	0.019178375	0.39201733
sisney2018	-15.688804	145.608487	NA	NA	NA	Undetermined	0.009127546	0.135568385	0.06732798
sussman2006	7.356796	134.423714	31/01/2004	31/01/2004	0	<4	0.00143815	0.074899242	0.01920113
thinesh2013	9.241999	79.2357	31/05/2008	31/05/2009	0.46	<4	0.643603389	0.062989299	10.2176628
viehman2006AR	25.14667	-80.29333	30/09/1999	30/09/2000	1.71	<4	0.111252753	0.000759836	146.416794

viehman2006PR	25.356932	-80.153403	30/09/1999	30/09/2000	1.52	<4	0.10228806	0.001072937	95.3346024
viehman2006TP	15.399919	-61.436267	30/09/1999	30/09/2000	1.75	<4	0.015151726	0.001286541	11.7771063
voss2006	23.77167	-76.0925	31/07/2003	31/07/2004	0	<4	0.01091445	0.002197115	4.96762705
voss2006GS	23.900315	-76.263181	31/07/2002	31/07/2002	0	<4	0.016866522	0.002055989	8.20360661
voss2006GS03	23.900315	-76.263181	31/07/2002	31/07/2002	0	<4	0.016866522	0.002055989	8.20360661
voss2006HS	23.77167	-76.0925	31/07/2002	31/07/2002	0	<4	0.01091445	0.002197115	4.96762705
voss2006HS03	23.77167	-76.0925	31/07/2002	31/07/2002	0	<4	0.01091445	0.002197115	4.96762705
voss2006NN	23.789042	-76.137413	31/07/2002	31/07/2002	0	<4	0.018845061	0.001855537	10.1561222
voss2006NN03	23.789042	-76.137413	31/07/2002	31/07/2002	0	<4	0.018845061	0.001855537	10.1561222
voss2006NP	23.780029	-76.099606	31/07/2002	31/07/2002	0	<4	0.01091445	0.002197115	4.96762705
voss2006NP03	23.780029	-76.099606	31/07/2002	31/07/2002	0	<4	0.01091445	0.002197115	4.96762705
voss2006RG	23.80169	-76.164775	31/07/2002	31/07/2002	0	<4	0.022166277	0.00172639	12.8396721
voss2006RG03	23.80169	-76.164775	31/07/2002	31/07/2002	0	<4	0.022166277	0.00172639	12.8396721
voss2006SP	23.774547	-76.095579	31/07/2002	31/07/2002	0	<4	0.01091445	0.002197115	4.96762705
voss2006SP03	23.774547	-76.095579	31/07/2002	31/07/2002	0	<4	0.01091445	0.002197115	4.96762705
voss2006SR	23.727518	-76.047868	31/07/2002	31/07/2002	0	<4	0.00741981	0.002349266	3.15835238
voss2006WH	23.794435	-76.127405	31/07/2002	31/07/2002	0	<4	0.018845061	0.001855537	10.1561222
voss2006WH03	23.794435	-76.127405	31/07/2002	31/07/2002	0	<4	0.018845061	0.001855537	10.1561222
voss2007BP	23.790007	-76.13871	31/07/2003	31/07/2004	0	<4	0.018845061	0.001855537	10.1561222
voss2007CS	25.210938	-80.219225	NA	NA	NA	Undetermined	0.108947802	0.000634916	171.593948
voss2007DR	25.125571	-80.296796	NA	NA	NA	Undetermined	0.111252753	0.000759836	146.416794
voss2007FR	25.037455	-80.350132	NA	NA	NA	Undetermined	0.132011904	0.000679685	194.225082
voss2007GR	25.111615	-80.305271	NA	NA	NA	Undetermined	0.114090421	0.000746218	152.891632
voss2007HB	18.349163	-64.678004	30/09/2004	30/09/2005	9.45	>8	0.002910709	0.001297928	2.24258041
voss2007HN	18.348612	-64.781199	30/09/2004	30/09/2005	9.2	>8			
voss2007MO	25.015008	-80.378541	NA	NA	NA	Undetermined	0.132011904	0.000679685	194.225082
voss2007NP	23.780029	-76.099606	31/07/2003	31/07/2004	0	<4	0.01091445	0.002197115	4.96762705

voss2007RB	23.80169	-76.164775	31/07/2003	31/07/2004	NA	Undetermined	0.022166277	0.00172639	12.8396721
voss2007SP	23.774547	-76.095579	31/07/2003	31/07/2004	0	<4	0.01091445	0.002197115	4.96762705
voss2007TB	23.758824	-76.114401	31/07/2003	31/07/2004	0	<4	0.018845061	0.001855537	10.1561222
voss2007WB	25.038181	-80.383034	NA	NA	NA	Undetermined	0.132011904	0.000679685	194.225082
voss2007WC	18.366576	-64.723605	30/09/2004	30/09/2005	9.26	>8	0.002886903	0.001256814	2.29700112
voss2007WH	23.794435	-76.127405	31/07/2003	31/07/2004	0	<4	0.018845061	0.001855537	10.1561222
wada2016	26.209563	127.271237	30/09/2022	30/09/2023	8.28	>8	0.089888938	0.029130509	3.08573174
wada2023AI14	26.209563	127.271237	31/08/2013	31/08/2014	3.78	<4	0.089888938	0.029130509	3.08573174
wada2023AI15	26.209563	127.271237	31/08/2014	31/08/2015	3.16	<4	0.089888938	0.029130509	3.08573174
wada2023SI14	26.653101	127.859077	31/08/2013	31/08/2014	4.05	4 to 8	0.131331957	0.032254861	4.07169485
wada2023SI15	26.653101	127.859077	31/08/2014	31/08/2015	2.09	<4	0.131331957	0.032254861	4.07169485
weil2009	32.358	-64.6415	31/12/2004	31/12/2005	0.15	<4	0.358978895	0.030324017	11.8381052
weil2012March	26.2985	127.721667	31/03/2009	31/03/2010	1.61	<4	0.087784213	0.025653592	3.42190719
weil2012Sept	26.2985	127.721667	30/09/2009	30/09/2010	1.61	<4	0.087784213	0.025653592	3.42190719
yang2014	16.848244	112.340973	30/04/2011	30/04/2012	1.05	<4	0.06830494	0.001952039	34.9915848
zimmer2014AR	25.14667	-80.29333	31/10/2008	31/10/2009	0.32	<4	0.111252753	0.000759836	146.416794
zimmer2014FL	26.133193	-80.088531	31/05/2007	31/05/2008	0.31	<4	0.092844197	0.003014866	30.7954655
zimmer2014FIHR	25.140329	-80.295458	30/09/2011	30/09/2012	1.56	<4	0.111252753	0.000759836	146.416794
zimmer2014WF	12.108623	-68.953608	28/02/2012	28/02/2013	5.29	4 to 8	0.020245312	0.010741401	1.88479257
zvuloni2009	29.49491	34.92387	31/07/2005	31/07/2006	0.8	<4	0.024551077	0.005397038	4.5489912

B

Supplementary Figure 3B Phylogeny of 16S rRNA amplicon sequences from *Thermodesulfobacterota* found in disease lesion bacterial mats ($n = 4$ disease lesion samples per nutrient condition). Hypothetical phylogenetic tree showing the relationship between 16S rRNA gene sequences identified in bacterial mats associated with tissue loss in *T. reniformis* and 16S rRNA gene sequences obtained from NCBI. A Maximum Likelihood phylogenetic tree of partial 16S rRNA gene sequences was constructed using IQ-TREE, with branch support estimated by 1,000 ultrafast bootstrap replicates. Spheres indicate the nutrient conditions from which each sequence was derived. Sequences obtained from NCBI for comparison contain their respective accession number and they are highlighted by a star. Clusters highlight groups containing closely related sequences identified in the BBD lesions from this study and NCBI-downloaded sequences obtained from BBD samples collected in distinct reef locations.